# Mechanical strain promotes skin fibrosis through LRG-1 induction mediated by ELK1 and ERK signalling

Ya Gao[1,4], Jia Zhou[1,4], Zhibo Xie[2,4], Jing Wang[3], Chia-kang Ho[1], Yifan Zhang[1]* & Qingfeng Li[1]*

Biomechanical force and pathological angiogenesis are dominant features in fibro-proliferative disorders. Understanding the role and regulation of the mechanical micro-environment in which pathological angiogenesis occurs is an important challenge when investigating numerous angiogenesis-related diseases. In skin fibrosis, dermal fibroblasts and vascular endothelial cells are integral to hypertrophic scar formation. However, few studies have been conducted to closely investigate their relationship. Here we show, that leucine-rich-alpha-2-glycoprotein 1 (LRG-1) a regulator of pathological angiogenesis, links bio-mechanical force to angiogenesis in skin fibrosis. We discover that LRG-1 is overexpressed in hypertrophic scar tissues, and that depletion of Lrg-1 in mouse skin causes mild neovascu-larization and skin fibrosis formation in a hypertrophic scarring model. Inhibition of FAK or ERK attenuates LRG-1 expression through the ELK1 transcription factor, which binds to the LRG-1 promoter region after transcription initiation by mechanical force. Using LRG-1 to uncouple mechanical force from angiogenesis may prove clinically successful in treating fibro-proliferative disorders.

[1] Department of Plastic & Reconstructive Surgery, Shanghai Ninth People's Hospital, School of Medicine, Shanghai Jiao Tong University, Shanghai, China. [2] Department of Pancreatic Surgery, Huashan Hospital, Shanghai Medical College, Fudan University, Shanghai, China. [3] Department of Otorhinolaryngology Head and Neck Surgery, Shanghai Children's Hospital, Shanghai Jiao Tong University, Shanghai, China. [4]These authors contributed equally: Ya Gao, Jia Zhou, Zhibo Xie. *email: zhangyifan82@126.com; dr.liqingfeng@shsmu.edu.cn

Hypertrophic scarring (HS) is a skin fibrotic disorder that occurs following extensive cutaneous injury with excessive fibrosis, characterized by aberrant fibroblast function[1], abundant collagen deposits[2], and superfluously formed microvessels[3]. HS not only leads to compromised cosmetic outcomes but also induces functional impairment[4]. Current therapeutic treatments for HS, including external medicine, steroid injection, cryotherapy, radiotherapy, and surgical operation, all exhibit limitations in yielding a satisfactory outcome[5–7]. Thus, it is urgent to uncover the molecular mechanisms of HS formation further to develop a promising treatment.

As previously reported, mechanical force is vital in the pathogenies of numerous fibro-proliferative diseases[8,9]. While the exact processes of hypertrophic scar formation are considered, mechanical force is also an indispensable component that must be examined[10–12]. Human dermal fibroblasts (HDFs) are mechano-sensitive cells that are integral to hypertrophic scar formation[13,14]. In HS, the taut and inelastic skin delivers tension to HDFs, which contributes to their over-activation and results in amplified collagen and fibronectin generation[15]. The excessive production of extracellular matrix (ECM) intensifies the stiffness of HS skin[16], leading to a persistent positive feedback loop that may result in the over-production of fibrotic matrix and subsequent fibrosis. Previous studies have reported that when mechanical stretching is sensed by cells, various types of mechano-sensitive elements, such as transmembrane receptors or ion channels, are activated including integrin-focal adhesion kinase (FAK) complex[11], stretch-activated ion channels[17], and G-Protein-coupled receptors[18]. Between these, the focal adhesion complex plays an essential role in linking the ECM and intracellular pathways. An earlier study also reported that FAK links mechanical force to skin fibrosis via an inflammatory signaling pathway[11]. It can activate downstream pro-fibrotic targets to transmit mechanical force and boost collagen production[19].

In addition, it is well-known that pathological angiogenesis is indispensable in hypertrophic scar formation[3]. The hypoxia environment in HS can induce further angiogenesis[20], which promotes cell proliferation, thus creating greater demand for oxygen. In the early stage of scar formation, recruited inflammation factors contributed to angiogenesis[21], while the newly formed microvessels exhibit endothelial dysfunction[22], leading to persistent inflammation. All aspects coalesce into a vicious circle of HS, and a crucial factor is pathological angiogenesis. Many angiogenesis factors like vascular endothelial growth factor (VEGF), transforming growth factor β1(TGF-β1), angiogenin, and thrombospondin were abnormal in HS[23]. Recently, a new modifier of pathological angiogenesis LRG-1 has been discovered[24]. LRG-1 is a highly conserved member of the leucine-rich repeat family of proteins; these proteins have been reported to be involved in cell adhesion, protein–protein interactions, and cell signaling[25–27]. A recent study revealed that LRG-1 modulates pathological angiogenesis by directly binding to the TGF-β accessory receptor endoglin[24].

Given that pathological angiogenesis plays an important role in HS formation and LRG-1 has been proved modulate neovascularization—mainly in the pathological situation—we aimed to explore whether LRG-1 constructed a bridge between biomechanical force and pathological angiogenesis, subsequently leading to HS formation. In the present study, we reveal that LRG-1 exhibit a high expression level in human and mouse hypertrophic scar tissues and an in vitro mechanical strain upregulates its expression in dermal fibroblasts. Mice lacking Lrg-1 develop mild neovascularization and skin fibrosis formation under mechanical force. Additionally, the signaling pathway that regulates LRG-1 expression during mechanical loading was uncovered. By manipulating LRG-1 expression, we may find a promising therapeutic treatment for HS and provide a new strategy for the treatment of diseases that involve biomechanical force and pathological angiogenesis, such as organ fibrosis and cancer.

## Results

**LRG-1 is overexpressed in human HS.** Firstly, we investigated the macromorphology and histology of normal human skin, atrophic scarring, and HS. As shown in Fig. 1a, HS skin exhibited a reddish appearance, suggesting it involves more pathological vessel formation. H&E staining demonstrated that there was a great change of dermal thickness and density in HS, while the neovascularization increased compared to normal skin and atrophic scarring (Fig. 1b). The immunohistochemical staining of endothelial cell marker CD31 confirmed an elevation of neovascularization in HS (Fig. 1c). Furthermore, the immunohistochemistry analysis revealed that LRG-1 is overexpressed in HS and was diffused in the dermis (Fig. 1d). Quantitative reverse transcription PCR (RT-qPCR) and Western blot analysis also showed that the mRNA and protein levels of LRG-1 were significantly higher in HS tissues (Fig. 1e, f). These results reflect our assumption that LRG-1 is associated with pathological angiogenesis in HS and scar hypertrophy.

**LRG-1 promotes HUVEC proliferation, migration, and angiogenesis.** To test whether LRG-1 plays a crucial role in angiogenesis, human umbilical vein endothelial cells (HUVECs) were incubated with exogenous LRG-1, and the biological effects were evaluated. The EdU proliferation assay revealed that the percentage of EdU-positive cells increased after 24 h of LRG-1 (300 and 500 nM) treatment (Fig. 2a). However, the apoptosis of HUVECs exhibited no significant difference (Fig. 2b). As the migration and angiogenesis capacity of HUVECs increased during neovascularization, the Transwell migration assay and Tube formation assay were conducted. The Transwell migration assay demonstrated that LRG-1 could increase the migration ability of HUVECs (Fig. 2c), and the Tube formation assay demonstrated the enhanced tube formation ability of HUVECs after the introduction of LRG-1 (Fig. 2d). It is known that HDFs play a major role in HS formation; the results highlight LRG-1's effect on HDFs. Our results indicate that there is no significant difference in HDFs' proliferation, apoptosis, migration, and contraction after addition of LRG-1. (Supplementary Fig. 1).

**AAV5-shRNA-mediated depletion of Lrg-1 attenuates load-induced hypertrophic scar formation in vivo.** To investigate whether the down-regulation of Lrg-1 in mouse skin can improve HS formation, a mechanical load-induced hypertrophic scarring model, which is histopathologically identical to human hypertrophic scarring, was employed[12]. Following the trend of human HS tissues, LRG-1 expression was significantly higher in mechanical load-induced mouse hypertrophic scar tissue than in control scar tissue (Fig. 3a). When mice with mechanical-load scarring were treated with AAV5-shLRG-1, the expression of LRG-1 was significantly down-regulated compared with AAV5-shCtrl-treated mice (Fig. 3b, c). Meanwhile, newly formed microvessels greatly decreased in the AAV5-shLRG-1 group according to the CD31 immunohistochemistry staining of CD31 and measurement of expression (Fig. 3d). After AAV5-shLRG-1 was administered, mice exhibited significantly decreased average scar area at each examined time point compared with AAV5-shCtrl-treated mice (Fig. 3e). Further histological analysis demonstrated that the cross-sectional size of the scar dramatically decreased in AAV5-shLRG-1-treated mice by day 14 (Fig. 3f). These results indicate that *LRG-1* knock-down hindered

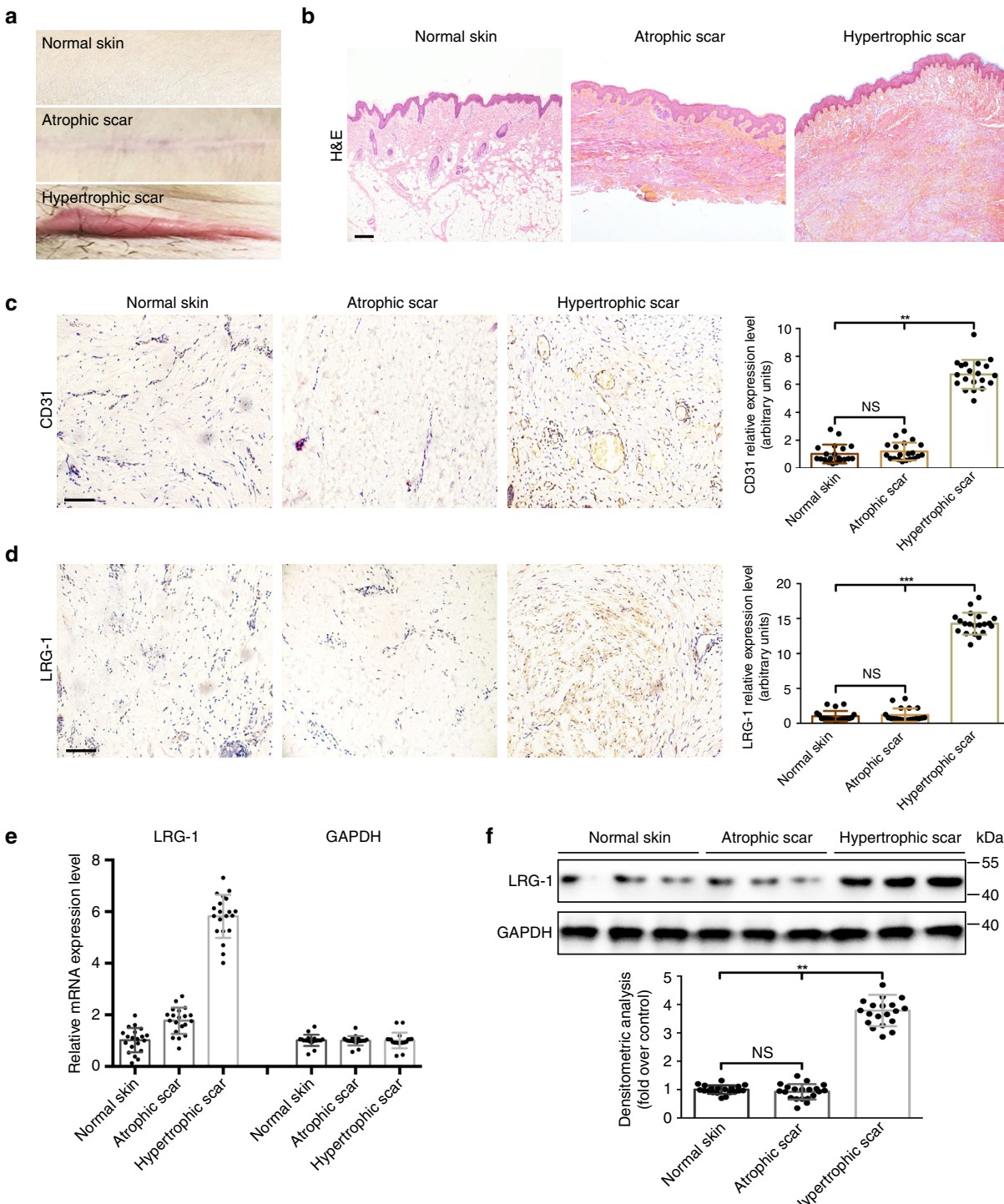

**Fig. 1** LRG-1 is overexpressed in human hypertrophic scarring. **a** Images of normal skin, atrophic scar, and hypertrophic scar. **b** Images of H&E-stained sections of normal skin, atrophic scar, and hypertrophic scar. (Scale bar = 200 μm). **c**, **d** Images and quantitative analysis of immunohistochemistry staining of CD31 and LRG-1. (Scale bar = 50 μm). **e**, **f** The levels of LRG-1 mRNA and protein in different skin tissues were measured using RT-qPCR and Western blotting. Data are presented as mean ± SD. n = 20 biologically independent samples. *P < 0.05, **P < 0.01, ***P < 0.001

pathological angiogenesis, thus attenuating load-induced hypertrophic scar formation in mice.

**LRG-1 is generated by HDFs due to mechanical loading**. We found that LRG-1 is a crucial protein in HS formation; thus, we were curious about how this protein was generated. The immunohistochemistry assay revealed that LRG-1 is primarily expressed in the fibrosis node, which is mostly composed of

myofibroblasts with high α-SMA expression (Fig. 4a). Meanwhile, as LRG-1 plays a vital role in angiogenesis, endothelial cells were also investigated to explore whether they could interact with LRG-1 to promote neovascularization. Considering that TGF-β1 has been proved plays a central role in HS formation[28],we were wondering whether the expression of LRG-1 in HS tissue was due to the effect of TGF-β1. However, our results demonstrated that LRG-1 expression was not obviously affected by TGF-β1 (Fig. 4b).

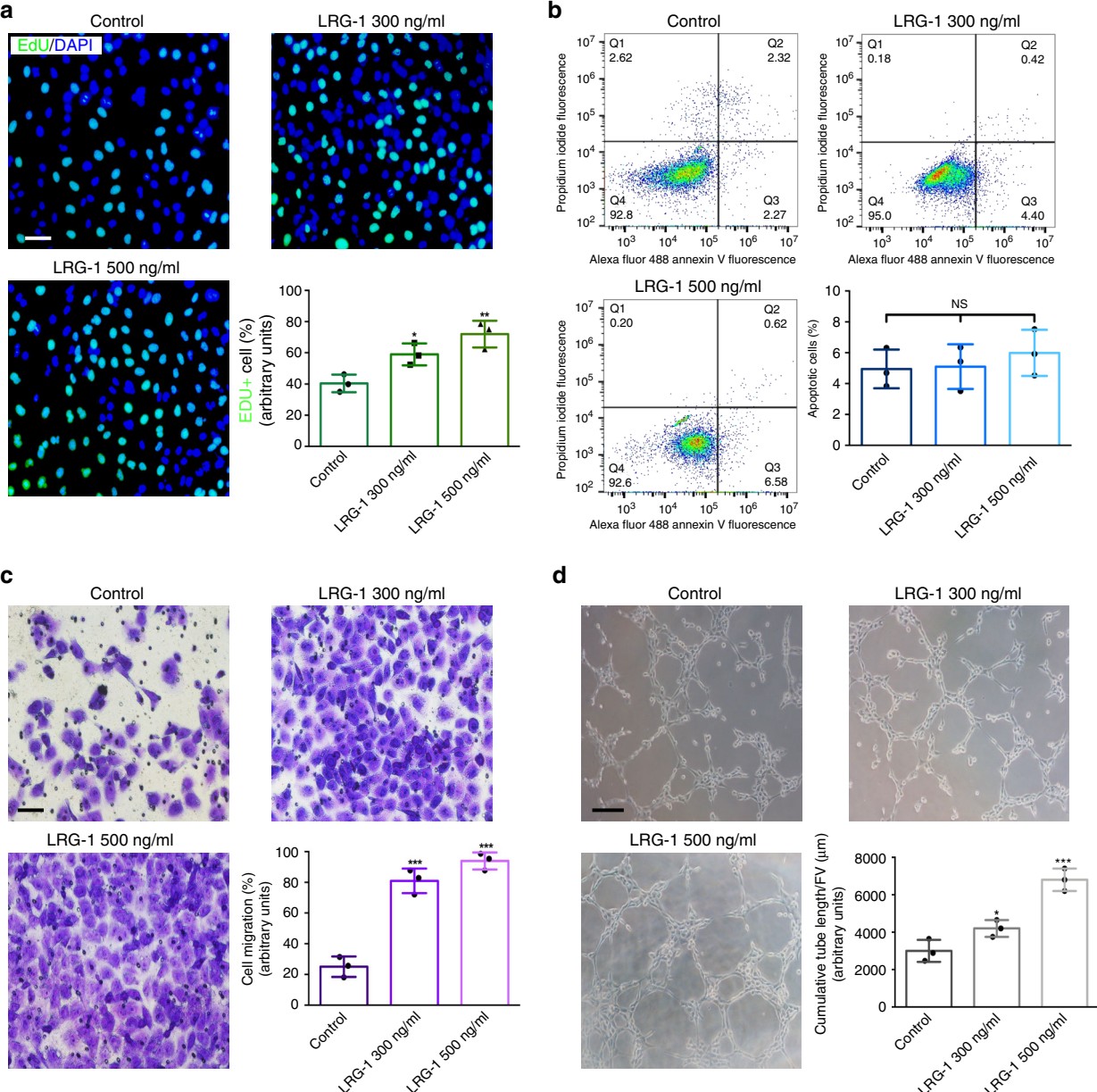

**Fig. 2** LRG-1 promotes HUVECs proliferation, migration, and angiogenesis. **a** EdU (green) proliferation assay was performed 24 h after the addition of 300 ng/mL and 500 ng/mL LRG-1 and the control group. DAPI-stained nuclei are blue. (Scale bar = 50 μm). **b** Apoptosis was detected after treating HUVECs with LRG-1 for 3 days by flow cytometry. **c** Transwell assay images and quantitative analysis for HUVEC migration after incubating with different concentrations of LRG-1. (Scale bar = 50 μm). **d** Matrigel tube-formation assay images and quantitative analysis of cumulative tube length. (Scale bar = 100 μm). Results were represented as means with standard errors (*n* = 3 independent experiments). *$P < 0.05$, **$P < 0.01$, ***$P < 0.001$

On the other hand, exaggerated inflammation has been shown to be one of the main mechanisms of excessive skin fibrosis[29]. Hence, we next investigated whether inflammation took part in the stimulation of LRG-1 generation. According to our in vitro inflammatory model induced by LPS[30], LRG-1 expression showed no obvious change in LPS-incubated HDFs and HUVECs (Fig. 4c). Furthermore, as mechanical loading has been revealed to be crucial for HS formation recently[11], we developed a cyclic mechanical strain to mimic the increasing stiffness environment *in vivo* during HS formation. ANKRD1 was used as a mechanical sensitive gene to confirm the mechanical loading environment (Fig. 4d). Our results demonstrated that mechanical loading significantly increased LRG-1 expression in a time- and strength-dependent manner in HDFs (Fig. 4e, f), while the expression level of LRG-1 in HUVECs remained low and unchanged (Fig. 4e, f).

These results suggest that mechanical force, rather than TGF-β1 or inflammation sensed by HDFs, triggered the over-expression of LRG-1.

**FAK/ERK signaling is critical for mechanical loading-induced LRG-1 expression**. To further investigate the underlying mechanism of mechanical stress–induced LRG-1 expression, FAK, a decisive element in mechanotransduction[11], was investigated. Our result showed that mechanical loading significantly up-regulated FAK phosphorylation in a time-dependent manner (Fig. 5a). When the FAK inhibitor was used, the up-regulation of protein levels of LRG-1 induced by mechanical loading was remarkably blocked (Fig. 5b). Next, we explored the downstream signaling pathway of FAK. We found that when HDFs were

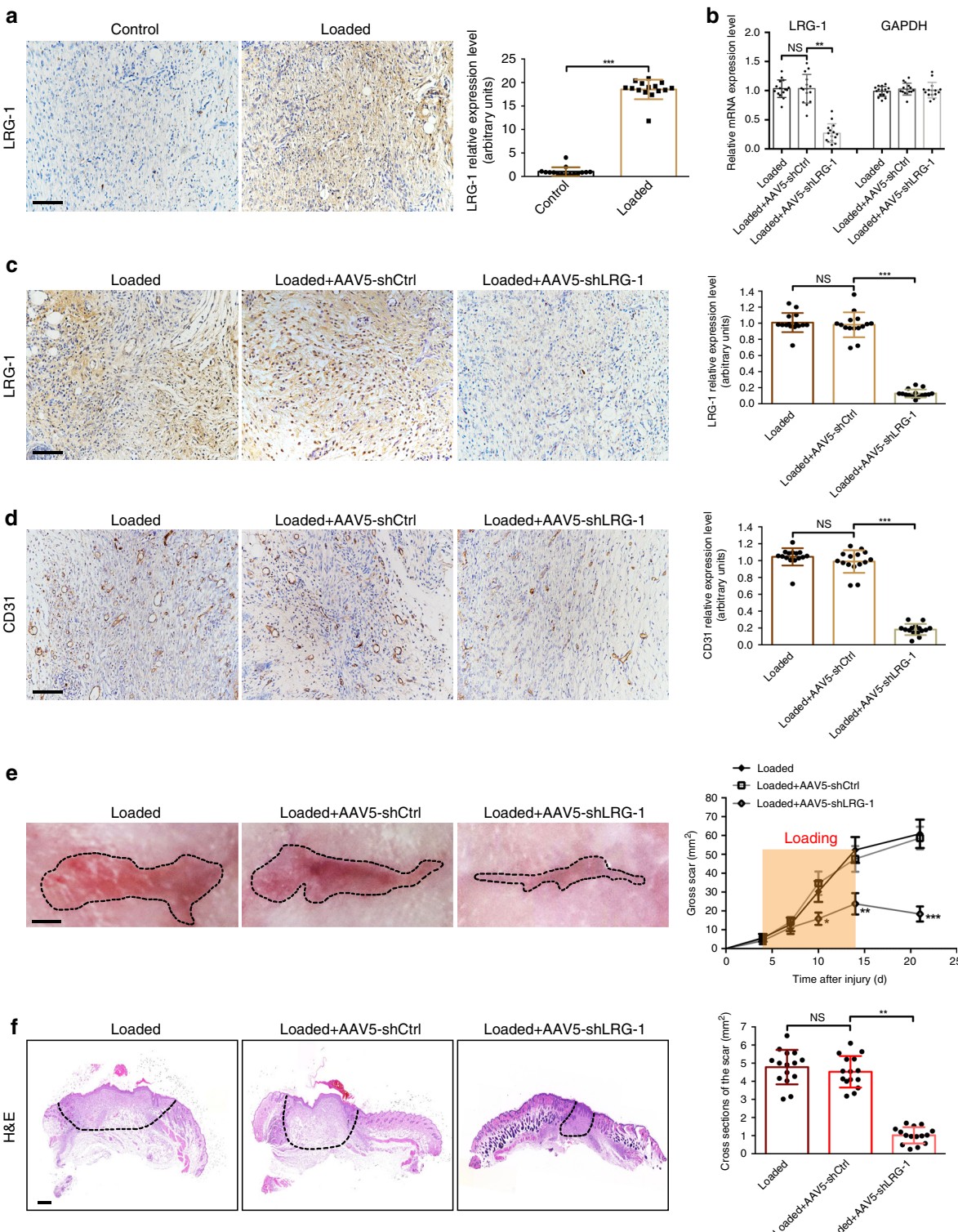

**Fig. 3** LRG-1 knock-down inhibits scar formation in a mechanic loading-induced mouse model. **a** Immunohistochemistry staining for LRG-1 in mouse scar tissues and expression level quantification. (Scale bar = 50 μm). **b** mRNA level of mouse skin of LRG-1 in loading group, loading with AAV5-shCtrl injection group and loading with AAV5-shLRG-1 injection group. **c, d** Immunohistochemistry staining for LRG-1 and CD31 of mouse scar tissues in three groups mentioned above. (Scale bar = 50 μm). **e** Gross pathology of scar tissue in three groups and gross scar areas quantification. The dashed lines outline the scar. (Scale bar = 3 mm). **f** Images of H&E stained sections and cross section size quantification. The dashed lines outline the scar. (Scale bar = 500 μm). Data are presented as mean ± SD. n = 15 biologically independent animals. *P < 0.05, **P < 0.01, ***P < 0.001

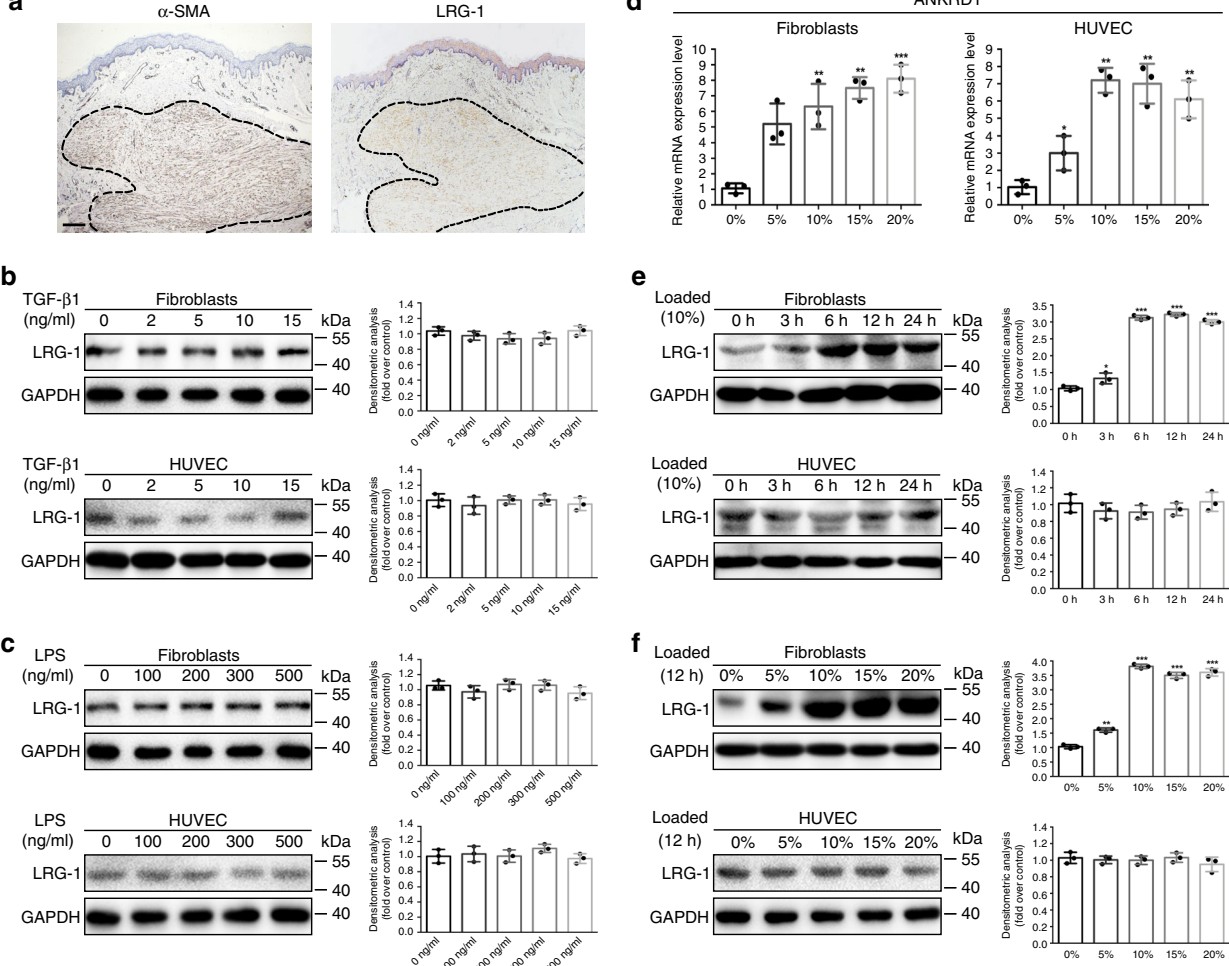

**Fig. 4** Mechanical stretch—not inflammation or TGF-β1—induces LRG-1 overexpression in human dermal fibroblasts (HDFs). **a** Immunohistochemistry staining images of α-SMA and LRG-1. (Scale bar = 200 μm). **b** LRG-1 levels were examined by Western blotting after HDFs and HUVECs were incubated with TGF-β1 in different concentration for 12 h. **c** LRG-1 expression after HDFs and HUVECs were incubated with LPS at different concentrations for 12 h. **d** mRNA expression of ANKRD1 were examined by RT-qPCR after HDFs and HUVECs were subjected to mechanical stretching at different strengths for 12 h. **e** LRG-1 expression after HDFs and HUVECs were subjected to mechanical stretch (10%) over different periods of time. **f** LRG-1 expressions after HDFs and HUVECs were subjected to mechanical stretch (12 h) at different strengths. Results were represented as means with standard errors (n = 3). *P < 0.05, **P < 0.01, ***P < 0.001

applied with 10% mechanical loading, the protein level of p-ERK markedly increased, whereas p-p38 and p-JNK showed no obvious changes (Fig. 5c). Additionally, ERK inhibition (but not JNK or p38) mostly diminished strain-induced LRG-1 expression (Fig. 5d). To further confirm that ERK activation was mediated by FAK, an FAK inhibitor was used during mechanical loading (Fig. 5e). Moreover, as shown by immunofluorescence, strain-induced translocation of ERK into the nucleus was performed, and LRG-1 expression was significantly decreased by both FAK and ERK inhibitor application (Fig. 5f). Consistent with our *in vitro* study, human HS tissues and mechanical load-induced mouse HS tissues exhibited higher p-FAK and p-ERK expression (Fig. 6a, b). All these data highlight the critical role of the FAK-ERK-LRG-1 axis in mechanotransduction in HDFs. Furthermore, HUVECs were also detected, and the results revealed that there was no change in p-ERK during mechanic loading (Supplementary Fig. 2).

Next, to confirm that FAK/ERK signaling is critical for mechanical loading-induced LRG-1 expression, FAK and ERK inhibitors were separately introduced in the mechanical load-induced hypertrophic scar model. As shown in Fig. 7a, b, sections

of FAK or ERK inhibitor-injected mouse scars exhibited significantly decreased LRG-1 and CD31 expression compared with the loading group. Moreover, we observed that the gross scar area and cross-sectional size were also significantly decreased in the two inhibitor-injected groups (Fig. 7c, d).

### ELK1 activation induced by FAK/ERK axis controls LRG-1 expression.
To understand the mechanism by which ERK regulates LRG-1 expression, we examined transcription factors (TFs) regulated by the ERK pathway, and we used PROMO and JAS-PAR to perform an online prediction of TF binding ability in the *LRG-1* promoter region (Fig. 8a, Supplementary Table 1); NFκB1 and ELK1 were the TFs of interest. To test whether these two TFs are involved in strain-induced LRG-1 production, we first conducted a Western blot assay, which demonstrated that ELK1 phosphorylation was stimulated but the p65 phosphorylation level remained unchanged by mechanical stress (Fig. 8b, Supplementary Fig. 3). Our Western blot analysis further demonstrated that the introduction of an FAK inhibitor decreased strain-induced ELK1 phosphorylation (Fig. 8c). As shown by

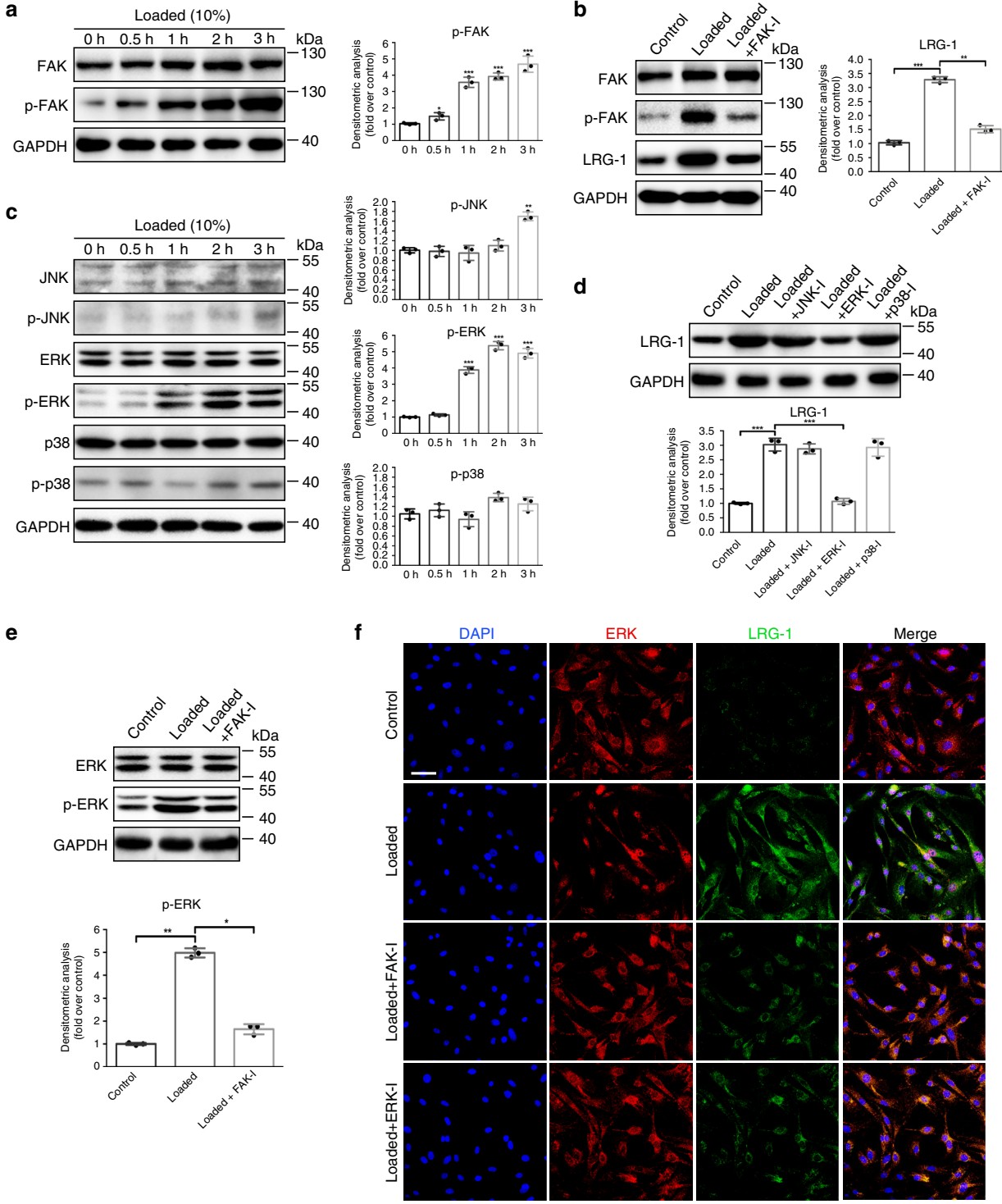

**Fig. 5** Mechanic loading induces LRG-1 expression via FAK-ERK signaling pathway. **a** Expression level of phosphorylated and total FAK after HDFs were subjected to mechanical stretching (10%) over different periods of time. **b** Western blotting analysis of LRG-1 and phosphorylation and total FAK in HDFs subjected to mechanical stretching (10%) and treated (or not treated) with FAK inhibitor (FAK-I) PF573228. **c** Expression level of phosphorylated and total JNK, ERK, and p38 after HDFs were subjected to mechanical stretching (10%) over different periods of time. **d** Mechanic loading-induced LRG-1 expression with small molecule inhibition of JNK (SP600125), ERK (PD98059), or p38 (SB203580). **e** Western blotting analysis of phosphorylation and total ERK in HDFs subjected to mechanical stretching (10%) and treated (or not treated) with FAK-I. **f** Immunofluorescence staining for ERK and LRG-1 in HDFs subjected to mechanical stretching (10%) and treated (or not treatment) with FAK-I or ERK-I. ERK are labeled in red and LRG-1 in green. (Scale bar = 50 μm). Results were represented as means with standard errors ($n = 3$ independent experiments). *$P < 0.05$, **$P < 0.01$, ***$P < 0.001$

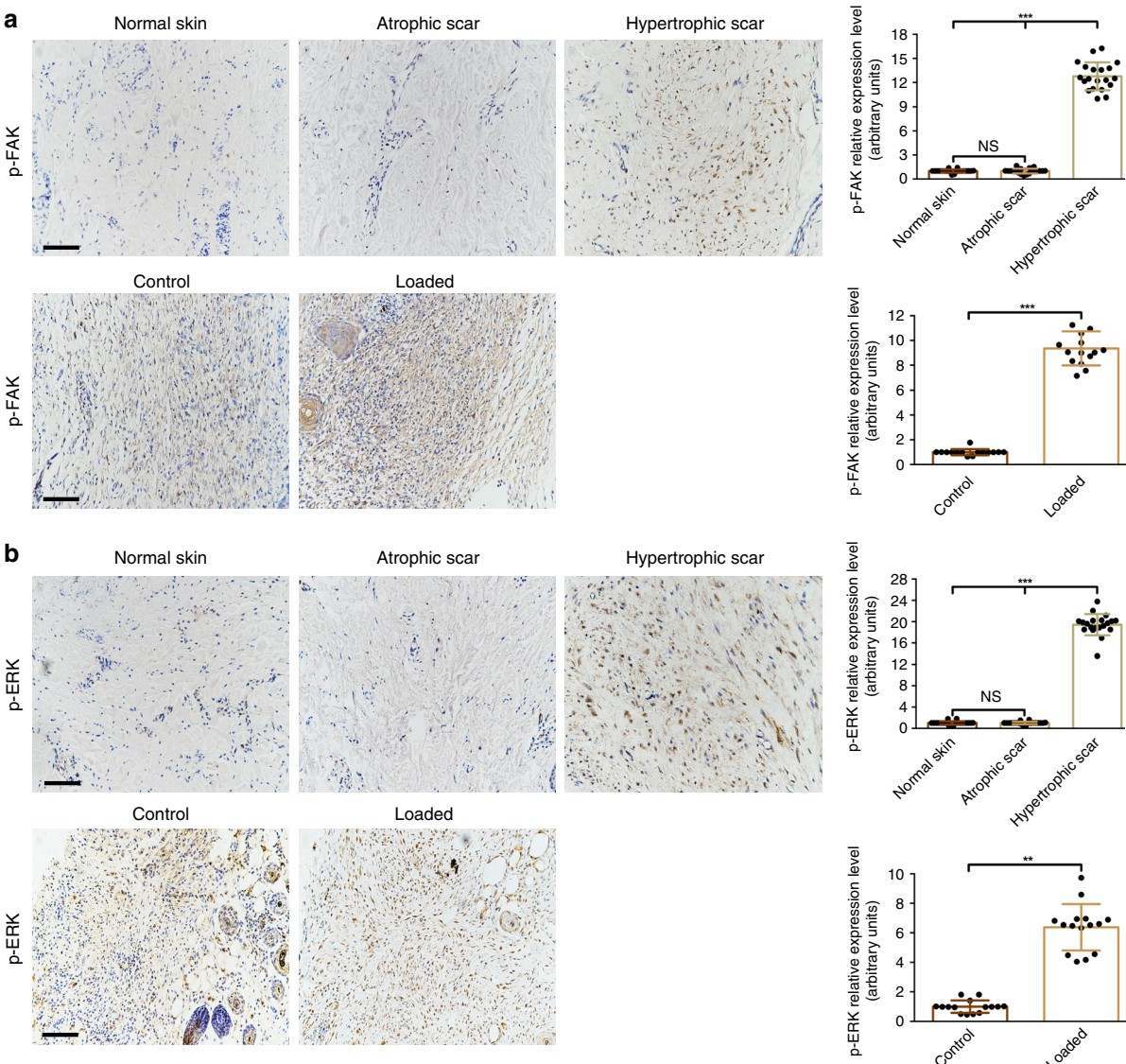

**Fig. 6** Immunohistochemistry staining of p-FAK (**a**) and p-ERK(**b**) in human skin tissues ($n = 20$) and mouse skin tissues ($n = 15$). (Scale bar = 50 μm). Data are presented as mean ± SD. **$P < 0.01$, ***$P < 0.001$

immunofluorescence, mechanical loading-induced translocation of ERK into the nucleus and ELK1 phosphorylation were significantly decreased by the application of an FAK inhibitor (Fig. 8d). Additionally, when the ERK inhibitor was used under loading conditions, ELK1 phosphorylation was also impeded (Fig. 8e). Next, we implemented siRNA experiments against ELK1 to inhibit its expression prior to mechanical stimulation (Fig. 8f). The Western blot and immunofluorescence assay both revealed that in siELK1 transfected–HDFs, strain-induced LRG-1 expression was mostly hindered (Fig. 8g, h). The luciferase activity reporter assay also proved that LRG-1 was a target gene of ELK1 (Fig. 8i). Moreover, our in vivo study demonstrated the same trend—ELK1 is over-activated in human HS tissues and mechanical load-induced mouse HS tissues (Fig. 8j). FAK or ERK inhibitor-injected mice presented markedly lower p-ELK1 expression compared with the purely loading group (Fig. 8j).

**ELK1 binds *LRG-1* promoter region**. To testify whether *LRG-1* is the direct target of ELK1, we mapped genome-wide ELK1 binding sites using Chromatin Immunoprecipitation Sequencing (ChIP-Seq) in HDFs under normal and loading conditions. Gene

ontology (GO) analyses and KEGG pathway analyses are shown in Fig. 9a and Supplementary Fig. 4B, C. We observed extensive binding at intergenic and intronic regions (Supplementary Fig. 4A); 15% of regulatory regions were located in gene promoters and enriched at transcription start sites (TSS, Fig. 9b). Furthermore, we can observe that the binding peaks at the TSS region exhibit significant differences between the control group and loading group (Fig. 9b, Supplementary Fig. 5). By combining the PROMO search results (five binding sites of ELK1 to *LRG-1* promoter regions) and the ChIP-seq results of differential distribution regions (chr19:4541300-4542400), with the ChIP certification followed by QPCR (ChIP-QPCR), we confirmed a binding site (chr19:4541670-4541678) of ELK1 to the *LRG-1* promoter region (Fig. 9c–e). Additionally, FAK or ERK inhibition was found to mostly diminish the strain-induced binding of ELK1 to the *LRG-1* promoter region (Fig. 9e).

## Discussion

In the present study, we discovered that LRG-1 plays a key role in the progression of skin fibrosis as well as a new mechanism linking biomechanical force and pathological angiogenesis.

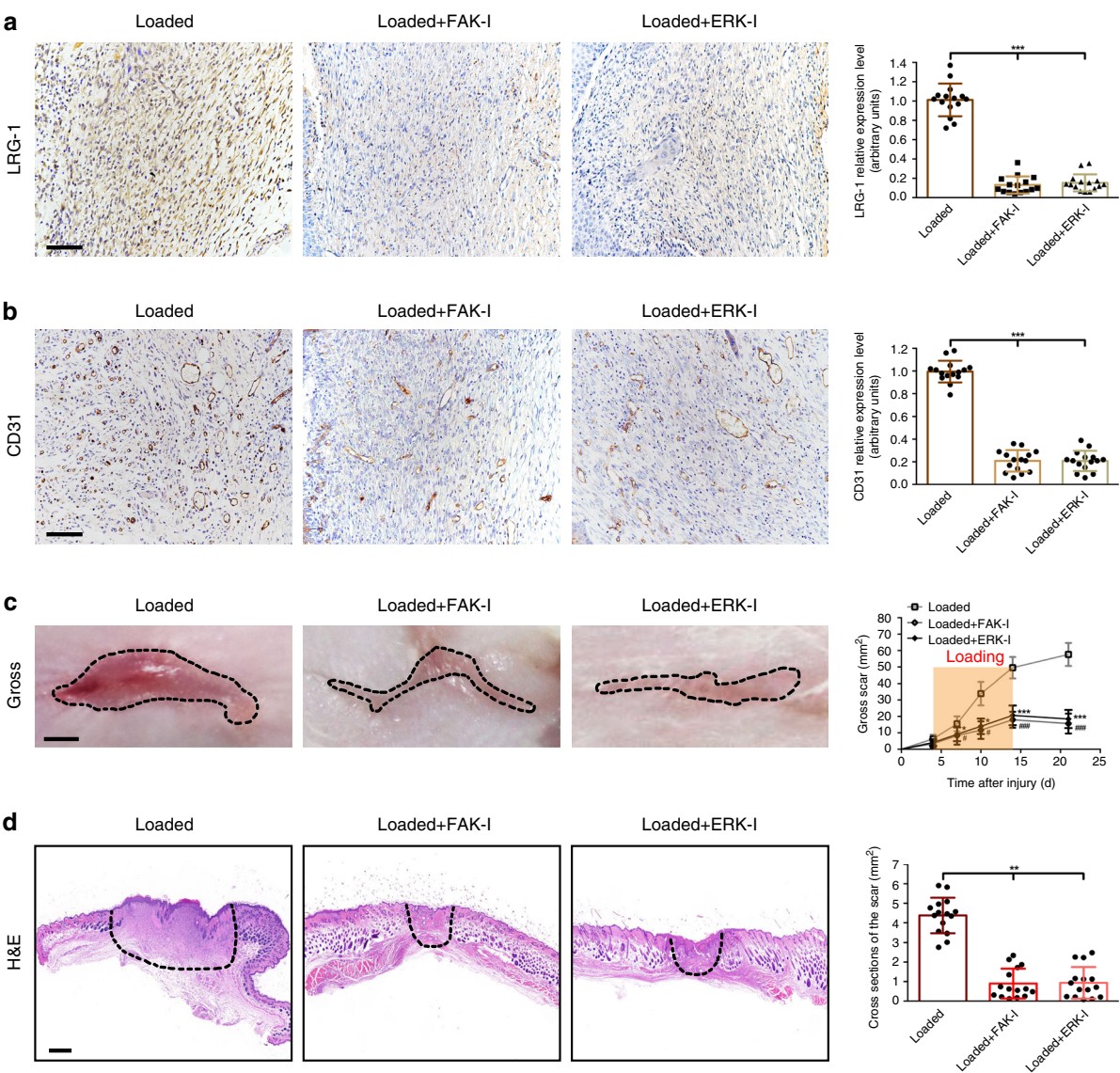

**Fig. 7** FAK or ERK inhibitor injection blocks mechanical loading-induced LRG-1 expression and attenuate scar formation. **a**, **b** Immunohistochemistry staining for LRG-1 and CD31 in mouse scar tissues of the loading group, loading with FAK inhibitor (PF573228) injection group, and loading with ERK inhibitor (PD98059) injection group and expression level quantification. (Scale bar = 50 μm). **c** Images of scars and gross area quantification at all examined time points. The dashed lines outline the scar. (Scale bar = 3 mm). **d** Images of H&E stained sections and cross-section size quantification. The dashed lines outline the scar. (Scale bar = 500 μm). Data are presented as mean ± SD. *n* = 15 biologically independent animals. *$P < 0.05$, **$P < 0.01$, ***$P < 0.001$

Specifically, during scar formation, mechanical loading delivered from ECM to the cell membranes of HDFs stimulates FAK phosphorylation, which then leads to the activation of ERK; p-ERK is transported from the cytoplasm to the nucleus and phosphorylates ELK1, resulting in co-factors binding with ELK1 at the promoter region of *LRG-1*, inciting the transcription process and stimulating LRG-1 expression. The overexpressed LRG-1 promotes pathological angiogenesis, subsequently aggravating the formation of a hypertrophic scar.

LRG-1, a secreted protein that galvanizes the TGF-β angiogenic switch, is speculated to play a more dominant role in disorganized pathological contexts than developmental/physiological angiogenesis[24]. Here, we revealed that mechanical force could induce pathological angiogenesis alone and that LRG-1 constructed a bridge between them. This finding contributes to further understanding the role and regulatory details of the mechanical microenvironment in which pathological angiogenesis occurs,

which is an important challenge in investigating numerous angiogenesis-related diseases, including fibro-proliferative disorders[31,32] and cancers[33]. Earlier studies have demonstrated that in organ fibrosis and malignant tumors, stroma exhibited fibrotic changes[34,35] such as ECM remodeling and collagen deposition[36,37]. Therefore, fibroblasts in cancerous or fibrotic tissue all are influenced by increasing mechanical stress generated by stiff ECM[38,39]. In the present study, we revealed that dermal fibroblasts, rather than endothelial cells, are mechanosensitive to mechanical stimuli and lead to the secretion of LRG-1 (Fig. 4), promoting pathological angiogenesis. Thus, fibroblasts in other fibrosis organs or tumor-associated fibroblasts under mechanical microenvironmental conditions may also be the main source of LRG-1 during these pathogenic processes. In line with the overexpression of LRG-1 in HS tissues observed in our study (Fig. 1d–f), cancerous liver tissue[40], endometrial carcinoma[41], and cancerous ovarian tissue[42] have been observed to express

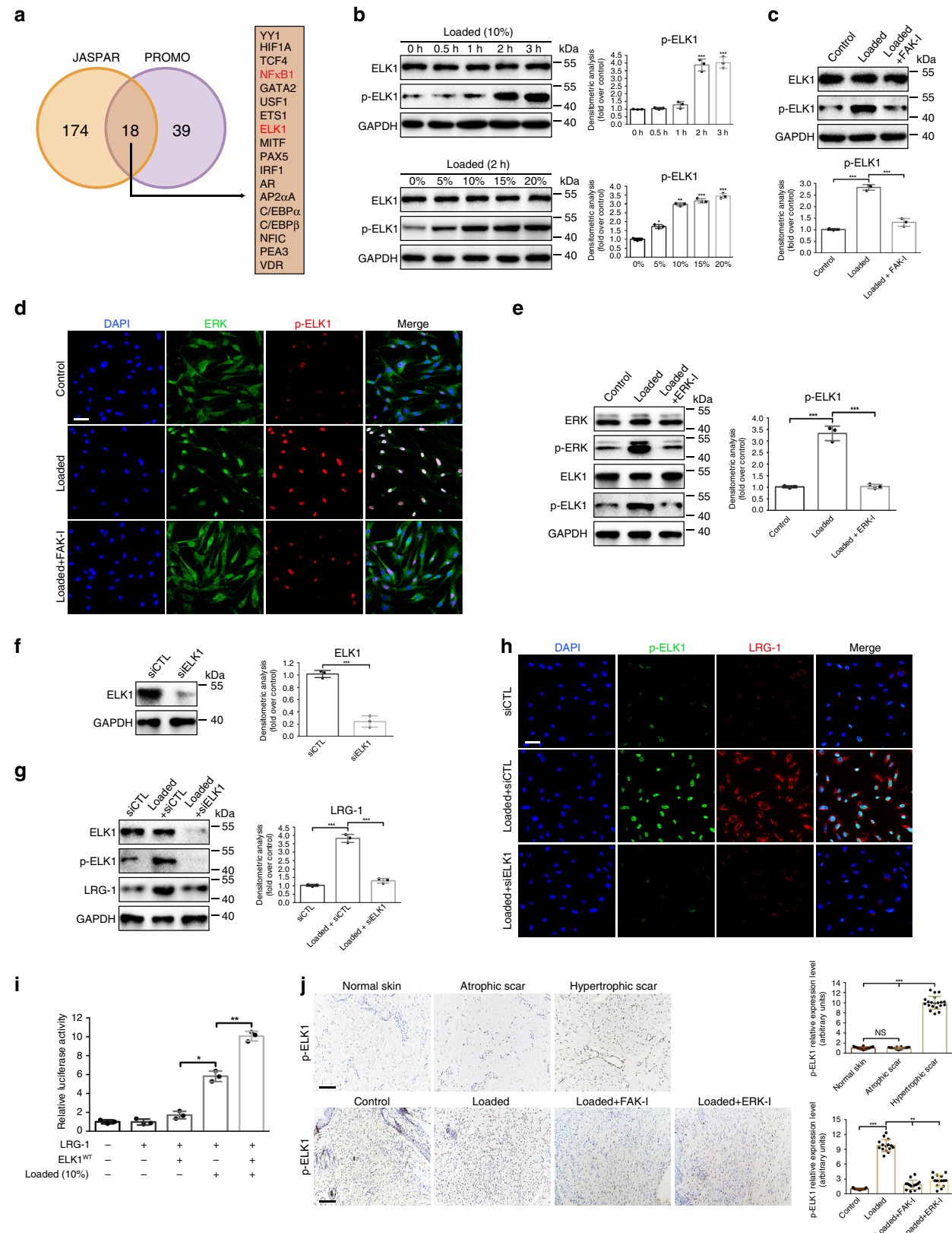

high levels of LRG-1. Additionally, we demonstrated that LRG-1 knock-down in mouse skin tissues remarkably attenuated HS formation in vivo (Fig. 3). Numerous angiogenesis-related diseases are combined with stiff ECM microenvironments[43,44]; interference of LRG-1 also may serve as a promising strategy for treating these disorders mentioned above.

Furthermore, a mechanism involved in the mechanical regulation of LRG-1 expression was revealed. Mechanical tension received by fibroblasts sensitized FAK, leading to ERK translocation to the nucleus, which results in ELK1 phosphorylation and enforces its reaction with the *LRG-1* promoter region (Fig. 9f). Previous studies have emphasized LRG-1's behavior in different

**Fig. 8** ELK1 controls LRG-1 expression. **a** Venn diagram showing the intersection of JASPAR and PROMO's online prediction of possible TFs that bind to the LRG1 promoter region. **b** Expression level of phosphorylated and total ELK1 after HDFs were applied after mechanical stretching (10%) at different periods of time or different strengths for 2 h, $n = 3$ independent experiments. **c** Western blotting analysis of phosphorylation and total ELK1 in HDFs applied with mechanical stretching (10%) and treatment (or no treatment) with FAK-I, $n = 3$. **d** Immunofluorescence staining for ERK and p-ELK1 in HDFs after mechanical stretching (10%) and treatment (or no treatment) with FAK-I. ERK are labeled in green and p-ELK1 in red, n = 3. (Scale bar = 50 μm). **e** Western blotting analysis of phosphorylation and total ERK and ELK1 in HDFs after mechanical stretching (10%) and treatment (or no treatment) with ERK-I, n = 3. **f** Protein expression levels of total ELK1 analyzed by Western blotting 2 days after siELK1 transfection, $n = 3$. **g** Effect of ELK1 inhibition on LRG-1 expression at day 3 after mechanical stretching (10%) (or not) and transfection at day 1 with siELK1 or siCTL, $n = 3$. **h** Immunofluorescence staining for p-ELK1 and LRG-1 in HDFs after mechanical stretching (10%) and transfection with siELK1 or siCTL. p-ELK1 are labeled in green and LRG-1 in red. (Scale bar = 50 μm). **i** Luciferase reporter assay demonstrates that LRG-1 is a target gene of ELK1, $n = 3$. **j** Immunohistochemistry staining of p-ELK1 in human skin tissues ($n = 20$, upper) and in mouse scar tissues of the control group, loading group, loading with FAK inhibitor (PF573228) injection group, and loading with ERK inhibitor (PD98059) injection group ($n = 15$, lower). (Scale bar = 50 μm). Data are presented as mean ± SD. *$P < 0.05$, **$P < 0.01$, ***$P < 0.001$

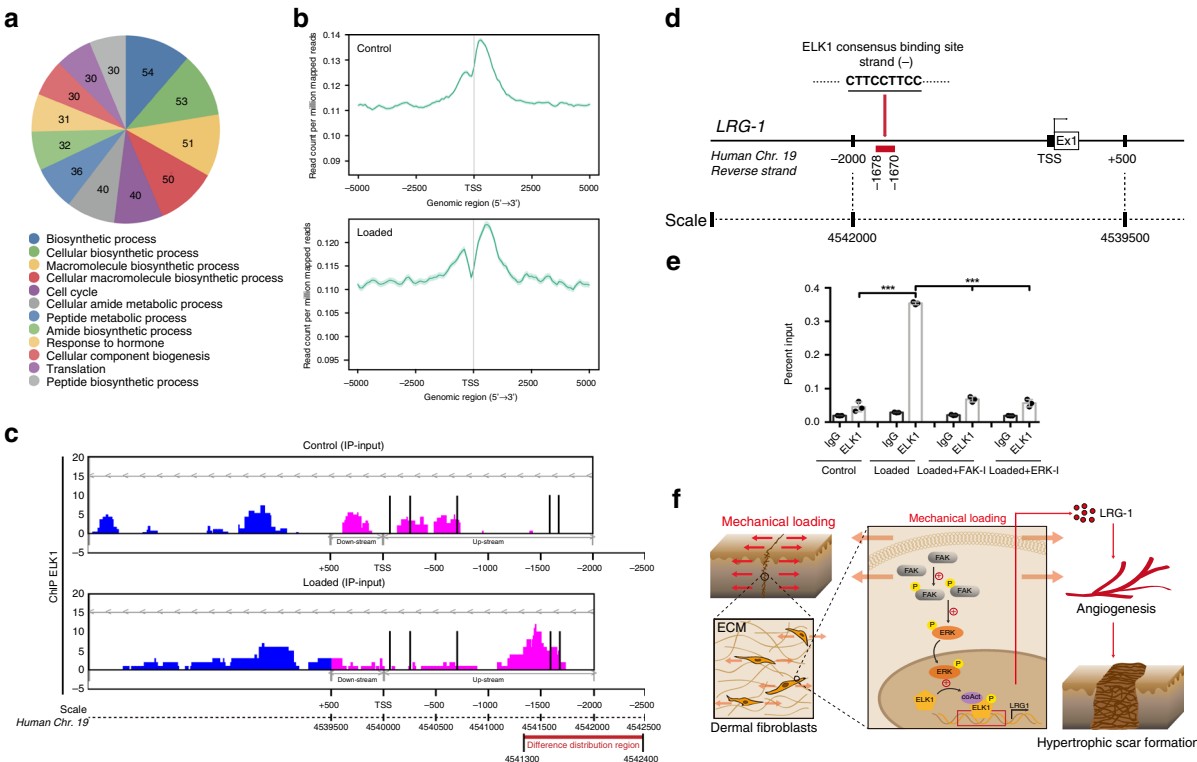

**Fig. 9** ELK1 binds to the *LRG-1* promoter region. **a** Functional analysis of upregulated genes of ChIP-seq results with DNA immunoprecipitated by ELK1 antibody. The numbers of genes in each functional category are presented. **b** Tag distribution (using bigWIG metrics) across transcription start sites (TSS): X-axis indicates the distance to TSS and Y-axis indicates read count per million mapped reads. **c** UCSC genome browser visualization of ChIP-seq data generated at control group and loading group in HDFs. ChIP-seq analysis reveals the main difference distribution peak in chr19:4541300-4542400. Five binding motifs of ELK1 in the *LRG-1* promoter region predicted by PROMO, marked with a black line. **d** Human *LRG-1* locus including the position of the ELK1 consensus binding site (PROMO prediction), within a promoter region (FANTOM5 prediction) and positions detected by RT-qPCR for the ChIP assays, marked with red boxes. Numbers indicate positions relative to the TSS (+1). Ex1 means Exon 1. **e** ChIP assay confirmation of the binding of ELK1 in control and loading groups of HDFs ($n = 3$ independent experiments). DNA immunoprecipitated by ELK1 antibody or immunoglobulin G (IgG CTL) was amplified by RT-qPCR using primers flanking the putative ELK1 binding site in predicted LRG-1 promoter position. **f** Proposed model of mechanical force promotes hypertrophic scar formation through FAK/ELK1 mediated LRG-1 expression. Data are presented as mean ± SD. ***$P < 0.001$

signaling pathways such as TGF-β[24] and HIF-1α[45]. Taking these results together with those from the present study, we have a more complete understanding of LRG-1's origin and function. Just as previous studies have demonstrated that FAK phosphorylation is an important mechano-transduction process that exerts great influence in tumor angiogenesis[46], tissue morphogenesis[47], and fibrosis[11], we concluded that FAK plays a vital role in mechanical-mediated pathological angiogenesis, which may occur in biological contexts not captured by our HS model. Besides some well-known mechano-response TFs such as β-Catenin[48], YAP[49], and AP-1[50], we characterized the mechanosensitive transcription factor ELK1 (Fig. 8) whose activity was regulated by

FAK/ERK along the mechanical loading process. Andrew E. et al. described that integrin-mediated organization of the actin cytoskeleton regulates localization of activated ERK and, in turn, the phosphorylation of ELK1[51]. Cytoskeleton remodeling is often accompanied with fine-tuned changes in cellular biomechanical conditions[52], this result is consistent with the observation in our mechanical model system. ELK1 is a transcription factor of the E-twenty-six (ETS) family at the crossroads of mitogen-activated protein kinase signaling cascades[53]. Different phosphorylation states and patterns of ELK1 exist and may vary with stimuli triggering ERK activation, thus determining transcriptional responses[54]. In this study, we explored the mechanically induced

ELK1 phosphorylation site at S383, which is involved in a broad range of molecular and cellular processes including cell mitosis[55], B cell differentiation[56], and epithelial-mesenchymal transition[57]. Thus, mechanical force may also participate in these processes through S383 phosphorylation of ELK1, which indicates a need for further investigation.

We revealed the association between mechanical force and pathological angiogenesis through our newly defined mechanical related secretory protein LRG-1. Moreover, our study identified the crucial role of LRG-1 in the progression of skin fibrosis. More essentially, we uncovered a detailed mechanism in which mechanical tension sensitized FAK, leading to ERK translocation into the nucleus, which results in the ELK1 phosphorylation and enforces its combination with *LRG-1* promoter region. These results demonstrate that LRG-1 is a promising therapeutic target for diseases that involve disturbed mechanical force and pathological angiogenesis, such as fibrogenic diseases and cancers.

## Methods

**Sample acquisition**. Twenty age- and site- matched normal skin tissues, atrophic scar tissues and hypertrophic scar tissues were obtained from Shanghai Ninth People's Hospital with ethics approval from local Human Research Ethics Committee of Shanghai Jiao Tong University School of Medicine in accordance with the Declaration of Helsinki principles. All samples were confirmed pathologically. Written informed consent was obtained from patients undergoing surgery to obtain excised tissue.

**Histology and immunohistochemistry**. Tissues which were paraformaldehyde-fixed overnight and then paraffin-embedded were cut at the thickness of 5 μm, then stained with hematoxylin and eosin (H&E) as previously reported[58]. As for immunohistochemistry, sections were incubated with primary antibody against LRG-1 (abcam, ab178698, 1:100) or LRG-1 (abcam, ab231188, 1:100) or CD31 (abcam, ab28364, 1:50) or α-SMA (abcam, ab5694, 1:100) or FAK (phosphoS732) (abcam, ab4792,1:100) or p44/42 MAPK antibody (Erk1/2) (CST, #4695, 1:50) or ELK1 (phospho S383) (abcam, ab218133, 1:100) dilution diluted in blocking solution overnight at 4 °C. After being incubated with HRP-conjugated secondary antibody, the sections were counterstained with hematoxylin and developed with diaminobenzidine.

Microvessel density was assessed through Chalkley method[59]. Briefly, in each section stained for CD31, the three most vascular areas were scored with a Chalkley eyepiece graticule. The Chalkley count for an individual section was taken as the cumulative value of the three graticule counts. For each sample, six randomly chosen sections from non-consecutive tissue sections were counted each time by two authors (Y.G. and Z.X.) independently and the intra-observer error was assessed. The microvessel density of a sample was taken as the mean Chalkley count of the six sections of the same sample.

**RNA purification and quantitative real-time PCR (RT-qPCR)**. The total RNA was isolated using TRIzol reagent (Invitrogen). RT-qPCR was performed with an ABI 7900HT system using SYBR Premix (Takara, Dalian, China) according to the manufacturer's instructions. Glyceraldehyde 3-phosphate dehydrogenase (GAPDH) was used as an internal control. The primers used in this study were as follows: GAPDH: forward, 5′-CCTCTGACTTCAACAGCGAC-3′; reverse, 5′-TCC TCTTGTGCTCTTGCTGG-3′; LRG-1: forward, 5′-GGACACCCTGGTATTGAA AGAAA-3′; reverse, 5′-TAGCCGTTCTAATTGCAGCGG-3′; ANKRD1: forward, 5′-AGTAGAGGAACTGGTCACTGG-3′; reverse, 5′-TGTTTCTCGCTTTTCCA CTGTT-3′.

**Western blotting**. Tissues and cultured cells were lysed with RIPA buffer supplied with protease inhibitor cocktail (Roche, Mannheim, Germany) as previously reported[58]. Concentrations of protein were detected by the bicinchoninic acid (BCA) assay (Thermo Fisher Scientific). To analyze inducible protein expression, 20 μg protein was resolved by 10% or 12% sodium dodecyl sulfate-polyacrylamide gel electrophoresis (SDS-PAGE) and electroblotted in the membranes of polyvinylidene difluoride (PVDF) (Millipore, Bedford, MA, USA). The membranes were blocked with 5% nonfat milk at room temperature for 1 h. The separated proteins were then immunoblotted, probed with primary anti-LRG-1 antibody (abcam, ab178698, 1:5000), anti-LRG-1 antibody (abcam, ab231188, 1:5000), anti-FAK antibody (abcam, ab40794,1:1000), anti-FAK (phosphoS732) antibody (abcam, ab4792,1:500), anti-p44/42 MAPK antibody (Erk1/2) (CST, #4695, 1:1000), anti-Phospho-p44/42 MAPK antibody (Erk1/2) (CST, #4370,1:2000), anti-SAPK/JNK antibody (CST, #9252, 1:1000), anti-Phospho-SAPK/JNK antibody (Thr183/Tyr185) (CST, #4668,1:1000), anti-p38 MAPK antibody (CST, #9212,1:1000), anti-Phospho-p38 MAPK antibody (Thr180/Tyr182) (CST, #4511, 1:1000), anti-ELK1 antibody (abcam, ab131465, 1:1000), anti-ELK1 (phosphoS383)

antibody (abcam, ab218133, 1:500), anti-GAPDH antibody (CST, #5174, 1:1000) at 4 °C overnight. Next day, the membranes were incubated with peroxidase-conjugated secondary antibody (1:1000) (Nebraska, USA) at room temperature for 1 h after washing with TBST 5 min for three times. Image J software was used for quantitative analysis which was conducted on immunoreactive bands. Full size western blot images are shown in Supplementary Fig. 7.

**Cell isolation and culture**. HUVECs were purchased from the ATCC (American Type Culture Collection). Primary human dermal fibroblast (HDFs) were isolated using the skin samples provided by Shanghai Ninth People's Hospital with ethics approval. Isolation steps are as fallows. After excision, use sterile 1 × PBS to wash the skin sample three times, then put it into 0.25% trypsin solution overnight at 4 °C. Next day use scissors to remove the epidermis, and cut dermal skin to small pieces. Then 0.25% Collagenase IV solution digest at 37 °C for 4 h. Passing through 200-mesh sieve. Centrifuge 5 min, 1000 r/min. HUVECs were cultured in Gibco™ RPMI 1640 and primary human dermal fibroblast (HDFs) were cultured in Gibco™ DMEM, high glucose, both supplemented with 10% fetal bovine serum and antibiotics (penicillin 100 IU/mL and streptomycin 100 mg/mL) in 5% $CO_2$ at 37 °C. For in vitro experiments, the doses of LRG-1 were used as 300–500 ng/mL according to previous study[24].

**EdU proliferation assay**. This procedure was conducted as previously reported[58]. To assess cell proliferation, cells were seeded in 24-well plates. The cells were incubated under standard conditions and were divided into three groups: control, 300 ng/mL LRG-1 added, 500 ng/mL LRG-1 added. 24 h after incubation, cell proliferation was detected by the incorporation of 5-ethynyl-2′-deoxyuridine (EdU) with the EdU Cell Proliferation Assay Kit (Invitrogen, Click-iT® EdU Imaging Kits). Steps were conducted according to the manufacturer's protocol. Briefly, the cells were incubated with 50 μM EdU for 2 h before fixation, permeabilization and EdU staining. Then cell nuclei were stained with DAPI (Sigma) at a concentration of 1 μg/mL for 8 min. The proportion of cells that incorporated EdU was determined by Zeiss 710 laser-scanning microscope (Zeiss, Thornwood, NY, USA).

**Apoptosis assay**. For apoptosis assay[58], LRG-1 pre-treated cells were re-suspended in PBS buffer by the amount of cells 5000/mL. In all, 195 μL cell suspension were mixed well with 5 μL Annexin V-FITC and incubated at room temperature for 10 min. Cells were washed with PBS and re-suspended in 190 μL deliquated binding buffer, then 10 μL 20 μg/mL PI were added. The samples were analyzed by flow cytometry using the Cell Quest program (BD Biosciences, San Jose, CA, U.S.A.).

**Migration assays**. This procedure was conducted as previously reported[58]. Transwell chambers containing polycarbonate filters in 8-mm pore size (Corning, Tewksbury, MA) were applied to evaluate the migration abilities of HUVECs and HDFs. Cells were cultured on the upper chamber with different concentration of LRG-1 in Gibco™ RPMI 1640 with no fetal bovine serum, and the lower chamber were filled with Gibco™ RPMI 1640 with 10% fetal bovine serum. The culture lasted for 48 h. Then the cells that migrated were fixed and stained for 30 min in a 0.1% Crystal Violet solution in PBS.

**Matrigel HUVECs tube formation assay**. HUVECs were grown on growth factor-reduced Matrigel (BD Biosciences). The 96-well plates were coated with Matrigel-containing culture medium (control) or LRG-1 in different concentration. Then allowed to polymerize in the incubator at 37 °C for 45 min. Tube formation was visualized using a stereo-microscope and analyzed by counting the number of branch points and total tube length per well using Image J.

**Collagen gel contraction assay**. HDFs were seeded into 24-well plates in 500 μL of collagen suspension (IBFB, Leipzig, Germany) and treated with different concentrations of LRG-1 (300 or 500 ng/mL) or DMSO. After collagen gel polymerization, the gels were released immediately from plates by tilting plates slightly. The area of each collagen gel was measured at day 3. Statistical analysis was done using Image J software.

**In vitro validation of the AAV-mediated gene silencing**. We utilized the AAV Helper-Free System (AAV Helper-Free System, Stratagene) for viral production using a triple-transfection, helper-free method, and purified as described in a previous study[60]. The interference sequences were as follows: 5′-GTCAGTGTGCA GATTCCTCAT-3′. Primary mouse dermal fibroblasts were infected with AAV vectors ($5 \times 10^9$ genome copies per 60 mm plate). 5 days after the infection with the AAV5-shLRG-1 or AAV5-shCtrl, RT-qPCR was performed and demonstrated that cells infected with AAV5-shLRG-1 resulted in a reduction of *LRG-1* mRNA levels by 75–77% relative to those transduced with AAV5-shCtrl.

**Animal ethics**. Animal welfare were strictly adhered to the principles of "Guide for the care and use of laboratory animals" (National Research Council. National

Academies Press; 27 December 2010). All procedures were performed in accordance with Guide for the Care and Use of Laboratory Animals which was approved by the Committee on the Ethics of Animal Experiments of Shanghai Jiao Tong University School of Medicine. The animals were housed in stable groups of three mice each in polycarbonate cages with autoclaved bedding. Each cage was provided with reverse-osmosis water delivered by an automatic water supply system and supplied with sterilized food. Room temperature was controlled by reheating units inside rooms and was maintained at $23 \pm 2\,°C$. The humidity was maintained at 30 to 70%. Animals were maintained on a 12:12-h light: dark cycle (lights on, 8 a.m. to 8 p.m.). At the end of in vivo experiment, euthanasia was conducted according to "CCAC guidelines on: euthanasia of animals used in science. Canadian Council on Animal Care". Body weight of mice were recorded from the beginning throughout the experiment everyday. Weight were plotted in Supplementary Fig. 6A.

**Animals and load-induced hypertrophic scar model**. C57BL/6 mice which were eight weeks old for the experiment were purchased from Shanghai Slac Laboratory Animal (Slac, Shanghai, China). The load-induced hypertrophic scar model was proceeded based on a model which was built by Geoffrey C Gurtner etc[12]. In brief, 1st day a 2 cm incision was made in the dorsal midline of the mice, then reapproximated with 6–0 nylon sutures. Fourth day, sutures were removed from the scars, and a specially made mechanical stretch devices were carefully secured with 6-0 nylon sutures. Mechanical load on the scars was created by carefully distracting the expansion screws in loading devices. The loading devices were distracted by 4 mm every other day to maintain the pressure and the stretch was maintained continuously from day 4th to 14th. Mechanical loading devices and the representative image of hypertrophic scar model were shown in Supplementary Fig. 6B. Mice were randomly grouped to four groups, including the control group, the loading group, the loading group with injection of AAV5-shCtrl, the loading group with injection of AAV5-shLRG-1. On day 14 (24 h after the last usage of AAV-virus), half the mice in each group were sacrificed for the sake of scar harvest and the other half was observed on day 21.

As for inhibitors injection, FAK inhibitor (PF573228, Selleck) was diluted using 5% DMSO + 2% Tween 80 + 30% PEG 300 + ddH$_2$O at the concentration of 5 mg/ml, ERK inhibitor (PD98059, Selleck) was diluted using 4% DMSO + 5% Tween 80 + 30% PEG 300 + ddH2O at the concentration of 1 mg/ml. The inhibitors injection protocols and time schedule were followed the procedures of AAV-virus.

**AAV vector administration**. Briefly, WT mice (8 weeks) were anaesthetized with an isofluorane/air mix (3% for initial induction and 1.5–2% for maintenance). Three hundred nanoliters of either AAV5-shLRG-1 or AAV5-shCtrl were injected into the subcutaneous of mice back skin 5 days before mechanic stretch and continued till the end of mechanic loading. The injections were performed using a 34-gauge needle (World Precision Instruments) attached to a 10 μL-NanoFil microsyringe (Nanofil, World Precision Instruments).

**Application of mechanical loading in vitro**. HDFs and HUVECs were plated at a density of $5 \times 10^5$ cells/cm$^2$ in 2 mL of medium on six-well flexible silicone rubber BioFlex plates coated with collagen type I (Flexcell International Corporation, Hillsborough, NC, USA). Cells were cultured for 24 h for adhesion and to reach 50–70% confluence before mechanical tension was applied. Cyclic mechanical stretch (CMS) with a 0.5-Hz sinusoidal curve at 10–20% elongation was applied using a Flexcell○RFX-5000$^{TM}$ Tension System (Flexcell○R International Corporation, NC, USA). The cultures were incubated in a humidified atmosphere at 37 °C and 5% CO$_2$ during the stretching. Cells were harvested immediately after the application of CMS stimulation was completed. Control cells were cultured on the same plates in the same incubator with no stretching[11].

**Immunofluorescence cell staining**. FAK inhibitor (PF573228, Selleck, S2013), ERK inhibitor (PD98059, Selleck, S1177) were used as recommended concentration during mechanic loading. SiRNAs (siELK1, siCTL) were pre-treated 48 h before the cyclic mechanical stretch applied. HDFs were fixed and blocked with 5% goat serum in PBST (0.1% Triton X-100 in phosphate-buffered saline) for 1 h. After washing, cells were incubated with primary antibodies against LRG-1 (abcam, ab178698) at 1:100 or against p44/42 MAPK antibody (Erk1/2) (CST, #4695) at 1:500 or against ELK1 (phosphoS383) (abcam, ab218133) at 1:100 for 2 h at room temperature, followed by the proper secondary antibody. Fluorescence was analysed using a Zeiss 710 laser-scanning microscope (Zeiss, Thornwood, NY, USA).

**siRNA and plasmid transfection**. For ELK1 silencing, HDFs were transfected in 6-well plates with 100 nM (final) ELK1 siRNA (sc-35290; Santa Cruz Biotechnology, Dallas, TX, USA) using Lipofectamine RNAiMAX reagent (Invitrogen, Carlsbad, CA, USA) according to the manufacturer's protocol. Nontargeting (NT) siRNA (sc-37007) was used as a negative control.

**Chromatin immunoprecipitation assay**. Chromatin immunoprecipitation (ChIP) was performed using Millipore Chip Kit (catalog #17-10085) and procedures were according to the manufacturer's protocol and a previously study[61]. Shortly

speaking, cells cultured under the previously indicated conditions were fixed in 1% formaldehyde/PBS for 10 min at room temperature. After two washes with PBS, cells were resuspended in 0.5 mL of lysis buffer containing a protease inhibitor cocktail before sonication. DNA fragments from the soluble chromatin preparations were 400–800 bp in length. Immunoprecipitation was carried out overnight with purified anti-ELK1(abcam, ab32106), anti-NFκB p65 (abcam, ab19870) or normal rabbit IgG as a negative control. Protein A/G agarose was used to pull down the antigen-antibody compounds and then washed four times with washing buffers. The DNA-protein crosslinks were reversed with 5 M NaCl at 65 °C for 6 h, and DNA from each sample was purified. PCR was performed using 2 μL DNA samples with the following primers: LRG-1 primer: forward, 5′-TGTCACTACA TTTCACAAGCCT-3′; reverse, 5′-CCAGCCGTTAGTTGGTCTTA-3′

**ChIP-seq**. Sequencing was performed on an Illumina HiSeq 2000 using TruSeq Rapid SBS Kits (Illumina, San Diego, CA, USA, FC-402-4002). The locations of ChIP-enriched DNA present in the library were based on the Human Feb 2009 assembly and visualized using the genome browser of the University of California. Peak calling in the mapped ChIP-Seq data was performed with ChIP-Peak and subjected to further bioinformatics analysis[61]. The ChIP-Seq procedures were performed by KangChen Biotech.

**Luciferase reporter assay**. Oligonucleotides used in the construction of the LRG-1 luciferase reporter plasmid was designed using Primer3. The reporter vector implemented for DNA insertion was the pGL3basic vector that encoded the Firefly luciferase reporter gene (Promega). The luciferase reporter containing the LRG-1 promoter (−1455; −967), pGL3 LRG-1-Luc, was constructed through Nested PCR amplification from HDFs DNA, followed by KpnI/Hind3 restriction digestion and ligation into the pGL3basic vector. The primers were as follows: the first primer: forward, 5′-GGGTTTCATCATATTGGTC-3′; reverse, 5′-GATGGAGTCTCCC TCTGC-3′; the second primer: forward, 5′-ATGAGGTACCGGGTTTCATCAT ATTGGTC-3′; reverse, 5′-TACGCTCGAGGGTTCAAACGATTCTCCTG-3′. pFA2-ELK1 expression plasmid was purchased from Beijing Huayueyang Biological Co., Ltd. (VECT 1236561).

Briefly, for the reporter assay, HDFs were plated at density of $8 \times 10^4$ cells per well in a 6-well flexible silicone rubber BioFlex plates 1 day before transfection. The pGL3 LRG-1-Luc and the pFA2-ELK1/pFA2-vector (control) were transfected using FuGENE 6 (Roche) according to the manufacturer's instructions. After 48 h transfection, cells were stretched at 10% elongation for 2 h. Then the cells were lysed and luciferase reporter activity was measured using the Luciferase Reporter system (Promega) with Firefly luciferase values normalized to Renilla luciferase values.

**Statistics and reproducibility**. Statistical software package SPSS 20.0 (SPSS, Chicago, IL, USA) was used for statistical analysis using the Mann−Whitney nonparametric test or the Student's $t$ test. All values were expressed as means ± standard deviation (SD). Analysis of samples was performed in triplicate and averaged. The difference between groups was regarded considerable at $P < 0.05$. All experiments were repeated at least three times.

**Reporting summary**. Further information on research design is available in the Nature Research Reporting Summary linked to this article.

## Data availability

All data supporting the findings of this study are available within the article and its Supplementary Information files, or are available on reasonable request from the corresponding authors. Source data are available as Supplementary Data 1. ChIP-seq data have been deposited in Gene Expression Omnibus with the primary accession code GSE119433.

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

## Acknowledgements

This study was supported by grants from National Natural Science Foundation of China (No. 81601688), Shanghai Sailing Program (grant No.19YF1426700) and National Natural Science Foundation of China (No. 81620108019).

## Author contributions

Conceptualization Y.Z. and Q.L.; methodology Y.G., J.Z., Z.X. and Y.Z.; investigation Y.G., J.Z., Z.X., J.W., C.H. and Y.Z.; analysis Y.G., J.Z., Z.X., X.H., Y.Z. and Q.L.; supervision Y.Z. and Q.L.; writing-original draft Y.G. and Y.Z.; review and editing Y.G., J.Z., Z.X., Y.Z. and Q.L.

## Competing interests

The authors declare no competing interests.
