## [Peer Review File · Communications Biology]

Reviewers' comments:

Reviewer #1 (Remarks to the Author):

Overall, this is a very nice manuscript that, with some modifications, would make an important contribution to the field.

Results:

The photomicrographs in Fig 1B,C and D are of poor quality. There is background staining for the IHC images which could be improved. PCR Fig 1F. Need to show control gene expression Fig 3B - again, need to show control gene expression

Line 144: active fibroblasts is not recognised nomenclature. α -SMA positive cells are termed myofibroblasts

Line 144. Localisation of LRG-1 staining to areas of fibrosis does not lead to the conclusion that dermal fibs are the major source.

Line 147. The authors need to explain the rationale for testing the effects of TGF- β 1 or LPS induced inflammation.

Fig 5G. Again, the resolution of the IHC images is low. Suggest that the authors show the IHC staining at a magnification such that cellular detail is apparent. If low power images are considered important, they can be depicted in supplementary data.

There are a number of spelling errors and some of the wording is difficult to follow. I appreciate that English may not be the authors' first language but it would be helpful to improve this before publication.

Reviewer #2 (Remarks to the Author):

The authors address a relevant question in the field of fibrosis, and identifies transcription factors of relevance.

The roles of FAK and ERK in fibrosis has been previously established; however, the present study identifies a new link between FAK-ERK and LRG-1 activation with subsequent promotion of angiogenesis.

The methodologies employed in this study are generally sound, with conclusions of some novelty.

However, there are several problematic areas the authors should address, potentially without the need for additional experiments.

(1) There is a lack of clarity on what cell types are over-expressing LRG-1 in Figure 1.

Overexpression of human LRG1 in endothelial cells has previously been shown to increase cell proliferation (Wang, et al., Nature, 2013). Is the LRG-1 over-expression presented in this study also found in endothelial cells? If so, why is LRG-1 protein added externally to HUVECs in Figure 2? Or is LRG-1 predominantly expressed by the fibroblasts, not endothelial cells? The logic behind these experimental designs needs to be clarified.

(2) A minor point about this statement: "mechano-sensitive elements in the cell membrane are activated like integrin-focal adhesion kinase (FAK) complex, stretch-activated ion channels and G-Protein-coupled receptors". - These are transmembrane receptors or ion channels that are mechano-sensitive. This is a minor issue where rewording would be recommended.

(3) The authors state: "This study providing a new sight that target LRG-1 to uncouple mechanical force from angiogenesis may prove clinically successful across diverse skin fibrosis or other fibroproliferative disorders."

This major conclusions relies on some in vitro studies with HUVECS and the findings presented in Figure 3D. However, the quantification method for Figure 3D is not clear. A major weakness of immunohistochemistry is its limited quantitative use. How representative are the images shown? Have non-consecutive tissue slides been used for the quantification? Most importantly, how exactly was the quantification performed? The authors have left this very vague and this is not acceptable in its current form.

Given the strength of the study relies heavily on the functional consequence of LRG-1 in relation to angiogenesis, the quantification methods need to be much more robust and transparent.

(4) There are typographical and grammatical errors throughout the manuscript that the authors should correct, otherwise the manuscript cannot be correctly interpreted by its readers. For example:

- LRG-1 has been proved modulate neovascularization
- LRG-1 constructed a bridge between biomechanical force and 84 pathological angiogenesis
- The Western blot and immunofluorescence assay 199 both turned out that in siELK1 transfected HDFs, strain-induced LRG-1 expression were 200 mostly hindered
- To testify whether LRG-1 plays a crucial role in angiogenesis.
- To understand the mechanism by which ERK regulates LRG-1 expression, we conducted researches on transcription factors (TFs) regulated by the ERK pathway, meanwhile we used

187 PROMO and JASPAR doing online prediction of TFs binding ability in LRG-1 promoter region.

- The Western blot and immunofluorescence assay both turned out that in siELK1 transfected HDFs, strain-induced LRG-1 expression were mostly hindered.
- Combine PROMO search results (5 binding sites of ELK1 to LRG-1 promoter region) and the ChIP-seq results of difference distribution region (chr19:4541300-4542400), with the certification by ChIP followed QPCR (ChIP-QPCR), we confirmed a binding site (chr19:4541670-4541678) of ELK1 to LRG-1 promoter region (Fig. 8C-E).

These are just some limited examples to demonstrate the need for the authors to check their grammatical errors in order to ensure the concepts they are trying to communicate are presented accurately.

(5) Animal ethics. Given the injurious nature of the in vivo experiments conducted, it would be desirable if the authors could provide additional animal welfare measurements to demonstrate appropriate welfare standards have been strictly adhered to. For example, body weight measurements at time points throughout the experiment would give an indication of the welfare standards that would be internationally acceptable.

Reviewer #3 (Remarks to the Author):

The manuscript focuses on the molecular mechanisms underlying neovascularization associated with hypertrophic scar formation. A strength of the research is the utilization of in vitro and animal models and the diverse tools used to address this issue. The authors demonstrate a pathway that includes focal adhesion kinase (FAK), ELK and LRG-1 in the response of fibroblasts to mechanical stimulation. The authors also demonstrate that LRG-1 acts on endothelial cells to promote neovascularization. In general, the experiments are well planned and the data are quite extensive. Overall the manuscript is well-written; however, substantial editing is needed to enhance the communication of the data and ideas in the paper.

Several suggestions include:

- Immunoblots should include size markers on the figures or in the figure legends.
- Rationale should be provided in the Methods section for the doses of LRG-1 used in the studies.
- Further explanation should be provided regarding the in vivo mechanical stimulation model. What was the frequency, magnitude and duration of the stimulation? Also, further discussion regarding the relevance of this model to hypertrophic scar formation would be helpful.
- The Y-axis of PCR data should indicate "Relative mRNA Expression"
- The X-axis of the graph in Figure 3E should state the units – is this hours?

Responses to Referees' comments:

Response to Reviewer 1:

“Overall, this is a very nice manuscript that, with some modifications, would make an important contribution to the field.”

Comment 1

The photomicrographs in Fig 1B,C and D are of poor quality. There is background staining for the IHC images which could be improved.

PCR Fig 1F. Need to show control gene expression

Fig 3B - again, need to show control gene expression

R: Thank the reviewer for the expert comments. We are sorry for the unbalanced background color of the photomicrographs in Fig. 1B, C and D which may due to the inappropriate white balance when they were acquired. We have readjusted white balance and revised these photomicrographs.

As for the second point of the comments, when you mentioned PCR, we assume you mean Fig. 1E? And according to this kind reminder, we have added control gene (GAPDH) expression in Fig. 1E and Fig. 3B.

Comment 2

Line 144: active fibroblasts is not recognised nomenclature. α -SMA positive cells are termed myofibroblasts

R: Thank the reviewer for pointing it out. The corrections have been made (under heading “LRG-1 is generated by HDFs due to mechanical loading”, paragraph 1).

Comment 3

Line 144. Localisation of LRG-1 staining to areas of fibrosis does not lead to the conclusion that dermal fibs are the major source.

R: Thank the reviewer for the expert comment and sorry for the inaccurate expression. The correlated sentence has been deleted (Given that, dermal fibroblasts may be the major cell type generate LRG-1).

Comment 4

Line 147. The authors need to explain the rationale for testing the effects of TGF- β 1 or LPS induced inflammation.

R: Thank the reviewer for his or her enlightening comment. In the process of organ fibrosis formation including skin fibrosis, TGF- β 1 has been proved plays a central role (Park SA et al., Cell Mol Life Sci. 2015) through stimulation of extracellular matrix (ECM) production, reactive oxygen species (ROS) generation, and myofibroblast activation (Leask A et al., FASEB J. 2004; Caraci F et al., Pharmacol Res. 2008; Crider BJ et al., J Invest Dermatol. 2011). Thus, as LRG-1 was overexpressed in human fibrotic skin tissue, we were wondering whether this phenomenon was due to the effect of TGF- β 1. However, our results demonstrated that LRG-1 expression was not obviously affected by TGF- β 1, which indicated that the overexpression of LRG-1 in hypertrophic scar was not

manipulated by TGF- β 1 (Fig. 4B).

On the other hand, exaggerated inflammation has been shown to be another main mechanisms of excessive skin fibrosis (Liu XJ et al., J Invest Dermatol. 2013). During skin scar formation, inflammatory cells such as neutrophils, macrophages and lymphocytes migrate to the wound site with high secretion of various cytokines, including TNF- α , IL-1 β and IL-6 (van der Veer WM et al., Burns. 2009; Ray S et al., J Invest Dermatol. 2013; Salgado RM et al., Burns. 2012), in which leading to fibroblast activation, massive collagen deposition and skin fibrosis formation. So far, gram negative bacterial lipopolysaccharide (LPS) is the most common way to induce inflammatory response. LPS could activate NF- κ B and MAPK pathways (Liu Y et al., Cell Death Differ. 2017) resulting in the expression of pro-inflammatory cytokines such as TNF- α , IL-1 β and IL-6 (Guo LT et al., J Neuroinflammation. 2019). Therefore, we investigated whether the expression of LRG-1 in scar tissue was induced by inflammation through LPS application. The results turned out that LPS had no effect on LRG-1 expression (Fig. 4C).

As we noticed the unclear expression in the corresponding section in our manuscript, we have revised the correlated description in the Results section according to the explanation above (under heading “LRG-1 is generated by HDFs due to mechanical loading”, paragraph 1). We thank the review again for introducing such a critical and constructive question.

Fig 5G. Again, the resolution of the IHC images is low. Suggest that the authors show the IHC staining at a magnification such that cellular detail is apparent. If low power images are considered important, they can be depicted in supplementary data.

R: Thank the reviewer for the expert comment. Since we aimed to present the comparison regarding the expression of p-FAK and p-ERK in normal skin and scar tissue, we chose images which are magnified 200 times. Under this amplification factor, those images not only contain larger field of vision but also can reflect the details of immunohistochemistry. After careful consideration of the reviewer's suggestions, we think perhaps it is because there were too many pictures placed in Fig. 5, resulting in a small size of each picture, which reduced the clarity of the pictures. We have separately listed the images in Fig. 5G and H as the new Fig. 6, and increased image size when exporting the images to get a clearer picture to reflect the details of immunohistochemistry.

Comment 6

There are a number of spelling errors and some of the wording is difficult to follow. I appreciate that English may not be the authors' first language but it would be helpful to improve this before publication.

R: Thank the reviewer for pointing this out. We have carefully reviewed the English and grammar and made the corrections.

Response to Reviewer 2:

“The authors address a relevant question in the field of fibrosis, and identifies transcription factors of relevance. The roles of FAK and ERK in fibrosis has been previously established; however, the present study identifies a new link between FAK-ERK and LRG-1 activation with subsequent promotion of angiogenesis. The methodologies employed in this study are generally sound, with conclusions of some novelty. However, there are several problematic areas the authors should address, potentially without the need for additional experiments.”

Comment 1

There is a lack of clarity on what cell types are over-expressing LRG-1 in Figure 1.

Overexpression of human LRG1 in endothelial cells has previously been shown to increase cell proliferation (Wang, et al., Nature, 2013). Is the LRG-1 over-expression presented in this study also found in endothelial cells? If so, why is LRG-1 protein added externally to HUVECs in Figure 2? Or is LRG-1 predominantly expressed by the fibroblasts, not endothelial cells?

The logic behind these experimental designs needs to be clarified.

R: Thank the reviewer for the expert comments. Indeed, we could not identify what cell types are over-expression LRG-1 in Fig. 1. In Fig. 1, we only aimed to illustrate that LRG-1 is overexpressed in human hypertrophic scar (HS) compared with the normal skin and atrophic scar.

As the reviewer mentioned Wang et al. (Wang X et al., Nature. 2013) has turned out that LRG-1 promotes angiogenesis by modulating endothelial TGF- β signaling. Hence, after we saw the rising trend of LRG-1 in HS, we did immunohistochemical staining of endothelial cells marker CD31 to testify whether there is an elevation of

neovascularization in HS accompanying the increased level of LRG-1 in HS (Fig. 1C). As the results in Fig. 1 showed that HS has both alleviated LRG-1 expression and neovascularization, we tested the impact of LRG-1 on HUVECs in Fig. 2 to confirm that LRG-1 could affect the biological behavior of endothelial cells. Consistent with *Wang et al's* study (Wang X et al., Nature. 2013), our results showed that LRG-1 could promote HUVECs proliferation, migration and angiogenesis. Furthermore, same as the reviewer considered, we were curious about how this protein was generated and wondered whether the LRG-1 over-expression presented in HS tissue in our study is derived from endothelial cells like Wang X et al (Wang X et al., Nature. 2013). Therefore, we further focused on which factors (TGF- β 1, inflammation and mechanical loading) are involved in the over-expression of LRG-1 and LRG-1 predominantly expressed by what kind of cells in HS. Our results demonstrated that LRG-1 expression was not obviously affected by TGF- β 1 and LPS in both human dermal fibroblast (HDFs) and endothelial cells (HUVECs) (Fig. 4B, C). However, mechanical loading significantly increased LRG-1 expression in HDFs in a time- and strength-dependent manner (Fig. 4E, F), while the expression level of LRG-1 in HUVECs stayed low and unchanged (Fig. 4E, F). These results indicated that LRG-1 predominantly expressed by the fibroblasts and mechanical force rather than TGF- β 1 or inflammation triggered the over-expression of LRG-1.

All of the above is our original logic behind these experimental designs, we wonder if we have explained the doubts of the reviewer, but we are willing to do more explanation if there are any doubts remains.

Comment 2

A minor point about this statement:

"mechano-sensitive elements in the cell membrane are activated like integrin-focal adhesion kinase (FAK) complex, stretch-activated ion channels and G-Protein-coupled receptors".

- These are transmembrane receptors or ion channels that are mechano-sensitive. This is a minor issue where rewording would be recommended.

R: Thank the reviewer for the kind reminder. According to this advice, we have changed our expression to "various types of mechano-sensitive elements, such as transmembrane receptors or ion channels, are activated including integrin-focal adhesion kinase (FAK) complex, stretch-activated ion channels, and G-Protein-coupled receptors." in our manuscript. The related information has been added to the Introduction section (under heading "Introduction", paragraph 2).

Comment 3

The authors state: "This study providing a new sight that target LRG-1 to uncouple mechanical force from angiogenesis may prove clinically successful across diverse skin fibrosis or other fibroproliferative disorders." This major conclusions relies on some in vitro studies with HUVECS and the findings presented in Figure 3D. However, the quantification method for Figure 3D is not clear. A major weakness of immunohistochemistry is its limited quantitative use. How representative are the images shown? Have non-consecutive tissue slides been used for the quantification? Most importantly, how exactly was the quantification performed?

The authors have left this very vague and this is not acceptable in its current form.

Given the strength of the study relies heavily on the functional consequence of LRG-1 in relation to angiogenesis, the quantification methods need to be much more robust and transparent.

R: Thank the reviewer for the expert comment and sorry for the unclarity. Microvessel density was assessed through Chalkley method (Chalkley HW, Journal of the National Cancer Institute. 1943; Fox SB et al., The Journal of pathology. 1995; Vermeulen PB et al., European journal of cancer. 2002).

Briefly, in each section stained for CD31, the three most vascular areas were scored with a Chalkley eyepiece graticule. The Chalkley count for an individual section was taken as the cumulative value of the three graticule counts. For each sample, six randomly chosen sections from non-consecutive tissue sections were counted each time by two authors (Ya Gao & Zhibo Xie) independently and the intra-observer error was assessed. The microvessel density of a sample was taken as the mean Chalkley count of the six sections of the same sample. Not only did we use this vascular density assessing method in Fig. 3D, we also used it in Fig. 1C and Fig. 7B. The images shown in figures are the representative pictures and the bar graphs are the statistical data. Chalkley method is widely used in microvessel density assessment (Suhonen KA et al., European Journal of Cancer. 2007; Waengertner LE et al., Gastroenterology research. 2011; Hansen S, et al., Histopathology. 2004) in pursuit of better quantification despite the major weakness of immunohistochemistry is its limited quantitative use as the reviewer mentioned. As Chalkley method has been proposed as a standard method of

microvasculature and is in general considered a simple and acceptable procedure for practical evaluation of vasculature, we think our conclusions drawn on this quantitative method are reliable. We want to thank the review again for pointing out our negligence, according to this kind reminder we have added the detail method of quantification in the Materials and Methods section (under heading "Histology and immunohistochemistry", paragraph 2).

Comment 4

There are typographical and grammatical errors throughout the manuscript that the authors should correct, otherwise the manuscript cannot be correctly interpreted by its readers. For example:

- LRG-1 has been proved modulate neovascularization*
- LRG-1 constructed a bridge between biomechanical force and 84 pathological angiogenesis*
- The Western blot and immunofluorescence assay 199 both turned out that in siELK1 transfected HDFs, strain-induced LRG-1 expression were 200 mostly hindered*
- To testify whether LRG-1 plays a crucial role in angiogenesis.*
- To understand the mechanism by which ERK regulates LRG-1 expression, we conducted researches on transcription factors (TFs) regulated by the ERK pathway, meanwhile we used 187 PROMO and JASPAR doing online prediction of TFs binding ability in LRG-1 promoter region.*
- The Western blot and immunofluorescence assay both turned out that in siELK1 transfected HDFs, strain-induced LRG-1 expression were mostly hindered.*

- Combine PROMO search results (5 binding sites of ELK1 to LRG-1 promoter region) and the ChIP-seq results of difference distribution region (chr19:4541300-4542400), with the certification by ChIP followed QPCR (ChIP-QPCR), we confirmed a binding site (chr19:4541670-4541678) of ELK1 to LRG-1 promoter region (Fig. 8C-E).

These are just some limited examples to demonstrate the need for the authors to check their grammatical errors in order to ensure the concepts they are trying to communicate are presented accurately.

R: Thank the reviewer for pointing this out and the kindly help. We have carefully reviewed the English and grammar and made the corrections.

Comment 5

Animal ethics. Given the injurious nature of the in vivo experiments conducted, it would be desirable if the authors could provide additional animal welfare measurements to demonstrate appropriate welfare standards have been strictly adhered to. For example, body weight measurements at time points throughout the experiment would give an indication of the welfare standards that would be internationally acceptable.

R: We thank the reviewer for this careful suggestion and kind reminder. Animal welfare were strictly adhered to the principles of “Guide for the care and use of laboratory animals” (National Research Council. National Academies Press; 2010 Dec 27). All procedures were performed in accordance with Guide for the Care and Use of Laboratory Animals which was approved by the Committee on the Ethics of Animal Experiments of Shanghai Jiao Tong University School of Medicine.

The animals were housed in stable groups of three mice each in polycarbonate cages with autoclaved bedding. Each cage was provided with reverse-osmosis water delivered by an automatic water supply system and supplied with sterilized food. Room temperature was controlled by reheating units inside rooms and was maintained at 23 ± 2 °C. The humidity was maintained at 30 to 70%. Animals were maintained on a 12:12-h light:dark cycle (lights on, 8 a.m. to 8 p.m.). At the end of *in vivo* experiment, euthanasia was conducted according to “CCAC guidelines on: euthanasia of animals used in science. Canadian Council on Animal Care.” (Charbonneau R et al., Ottawa ON, Canada. 2010) before skin sampling. Body weight measurements of mice were recorded from the beginning of the experiment everyday, and their body weight were plotted as curves and have been added as Supplementary Fig. S6A.

Thanks again for this reminder, and we have added Animal ethics part in our Materials and Methods section of the manuscript (under heading “Animal ethics”, paragraph 1).

Response to Reviewer 3:

“The manuscript focuses on the molecular mechanisms underlying neovascularization associated with hypertrophic scar formation. A strength of the research is the utilization of in vitro and animal models and the diverse tools used to address this issue. The authors demonstrate a pathway that includes focal adhesion kinase (FAK), ELK and LRG-1 in the response of fibroblasts to mechanical stimulation. The authors also demonstrate that LRG-1 acts on endothelial cells to promote neovascularization. In general, the experiments are well planned and the data are quite extensive. Overall the manuscript is well-written; however,

substantial editing is needed to enhance the communication of the data and ideas in the paper.

Several suggestions include: ”

Comment 1

Immunoblots should include size markers on the figures or in the figure legends.

R: Thank the reviewer for this useful suggestion. According to this advice, the original molecular weight markers have been added to the related figures.

Comment 2

Rationale should be provided in the Methods section for the doses of LRG-1 used in the studies.

R: We thank the reviewer for the comment and sorry for the unclarity. The dose of LRG-1 used in our study (300-500 ng/ml) was according to a previous study (Wang X et al., Nature. 2013). We have cited this reference (Wang X et al., Nature. 2013) to the Materials and Methods section (under heading “Cell isolation and culture”, paragraph 1).

Comment 3

Further explanation should be provided regarding the in vivo mechanical stimulation model.

What was the frequency, magnitude and duration of the stimulation? Also, further discussion regarding the relevance of this model to hypertrophic scar formation would be helpful.

R: Thank you for pointing it out and sorry for the unclarity. In our

mechanical-load-induced hypertrophic scar mouse model, mechanical load on the scars was created by carefully distracting the expansion screws in stretch devices. The stretch devices were distracted by 4 mm every other day to maintain the pressure and the stretch was maintained continuously from day 4th to 14th. We have added more details about mechanical load-induced hypertrophic scar model in the Material and Methods section (under heading “Animals and load-induced hypertrophic scar model”, paragraph 1). Meanwhile, we added the pictures of mechanical loading device and the hypertrophic scar model in Supplementary Fig. S6B.

Regarding the second point of the comments, this mechanical-load-induced hypertrophic scar model was designed by Gurtner GC and his colleague for studies of the relationship of mechanical-load and hypertrophic scar formation (Aarabi S et al., FASEB J. 2007). They demonstrated that the mechanical-load-induced scars in mouse are histopathologically identical to human hypertrophic scars. Nowadays, this kind of scar model is widely used for the study of hypertrophic scar (Wong VW et al., Nat Med. 2012; Zhang YF et al., Br J Dermatol. 2016). We have added correlated discussion in the Results section (under heading “AAV5-shRNA-mediated depletion of Lrg-1 attenuates load-induced hypertrophic scar formation *in vivo*”, paragraph 1).

Comment 4

The Y-axis of PCR data should indicate “Relative mRNA Expression”

R: We thank the reviewer for the comment and sorry for the unclarity. According to this suggestion, the Y-axis of PCR data have been revised.

Comment 5

The X-axis of the graph in Figure 3E should state the units – is this hours?

R: We thank the reviewer for the comment and sorry for the unclarity. The X-axis units of the graph in Fig. 3E is days. We have added the X-axis units of Fig. 3E and revised the Fig. 3.

In all, I found the reviewer's comments are quite helpful, and I revised my paper point-by-point.

Thank you and the reviewers again for your help!

REVIEWERS'

COMMENTS:

Reviewer #1 (Remarks to the Author):

The authors have extensively revised the manuscript. However, language remains a problem.

For example, I am unable to interpret the following in the abstract: "and that an in vitro mechanical strain was up-regulated in dermal fibroblasts."

Tidying up of the language is not simply a matter of style. As it stands, the authors risk the readers not understanding some of the points they make in the manuscript.

Fig 4A should show serial sections stained for α SMA and LRG-1

Reviewer #2 (Remarks to the Author):

The authors have taken sufficient steps to address the comments and questions raised. The quantification method for immunohistochemistry have been clarified and animal welfare standards have been addressed and added to supplemental information. The language has been improved greatly for clear communication.

The other points raised are also sufficiently addressed, and I would recommend publication at this stage.

Responses to Referees' comments:

Response to Reviewer 1:

The authors have extensively revised the manuscript. However, language remains a problem. For example, I am unable to interpret the following in the abstract: "and that an in vitro mechanical strain was up-regulated in dermal fibroblasts." Tidying up of the language is not simply a matter of style. As it stands, the authors risk the readers not understanding some of the points they make in the manuscript.

R: Thank the reviewer for pointing this out. We have carefully revised our language and grammar and made the corrections in our manuscript.

Fig 4A should show serial sections stained for α SMA and LRG-1

R: Thank the reviewer for the expert comments. According to this suggestion, we replaced Fig. 4A using the serial sections stained for α -SMA and LRG-1.

Response to Reviewer 2:

The authors have taken sufficient steps to address the comments and questions raised.

The quantification method for immunohistochemistry have been clarified and animal welfare standards have been addressed and added to supplemental information.

The language has been improved greatly for clear communication.

The other points raised are also sufficiently addressed, and I would recommend publication at this stage.

R: Thanks to the reviewer for helping us during the revision process.

In all, I found the editors' and reviewers' comments are quite helpful. Thank you and the reviewers again for your help!